# Learning to Learn without Forgetting By Maximizing Transfer and Minimizing Interference

Matthew Riemer[1,3], Ignacio Cases[2], Robert Ajemian[4,3], Miao Liu[1,3], Irina Rish[1,3], Yuhai Tu[1,3], and Gerald Tesauro[1,3]

[1]IBM Research, Yorktown Heights, NY
[2]Linguistics and Computer Science Departments, Stanford NLP Group, Stanford University
[3]MIT-IBM Watson AI Lab
[4]Department of Brain and Cognitive Sciences, MIT

## ABSTRACT

Lack of performance when it comes to continual learning over non-stationary distributions of data remains a major challenge in scaling neural network learning to more human realistic settings. In this work we propose a new conceptualization of the continual learning problem in terms of a temporally symmetric trade-off between transfer and interference that can be optimized by enforcing gradient alignment across examples. We then propose a new algorithm, Meta-Experience Replay (MER), that directly exploits this view by combining experience replay with optimization based meta-learning. This method learns parameters that make interference based on future gradients less likely and transfer based on future gradients more likely.[1] We conduct experiments across continual lifelong supervised learning benchmarks and non-stationary reinforcement learning environments demonstrating that our approach consistently outperforms recently proposed baselines for continual learning. Our experiments show that the gap between the performance of MER and baseline algorithms grows both as the environment gets more non-stationary and as the fraction of the total experiences stored gets smaller.

## 1 SOLVING THE CONTINUAL LEARNING PROBLEM

A long-held goal of AI is to build agents capable of operating autonomously for long periods. Such agents must incrementally learn and adapt to a changing environment while maintaining memories of what they have learned before, a setting known as lifelong learning (Thrun, 1994; 1996). In this paper we explore a variant called continual learning (Ring, 1994). In continual learning we assume that the learner is exposed to a sequence of tasks, where each task is a sequence of experiences from the same distribution (see Appendix A for details). We would like to develop a solution in this setting by discovering notions of tasks without supervision while learning incrementally after every experience. This is challenging because in standard offline single task and multi-task learning (Caruana, 1997) it is implicitly assumed that the data is drawn from an i.i.d. stationary distribution. Unfortunately, neural networks tend to struggle whenever this is not the case (Goodrich, 2015).

Over the years, solutions to the continual learning problem have been largely driven by prominent conceptualizations of the issues faced by neural networks. One popular view is catastrophic forgetting (interference) (McCloskey & Cohen, 1989), in which the primary concern is the lack of stability in neural networks, and the main solution is to limit the extent of weight sharing across experiences by focusing on preserving past knowledge (Kirkpatrick et al., 2017; Zenke et al., 2017; Lee et al., 2017). Another popular and more complex conceptualization is the stability-plasticity dilemma (Carpenter & Grossberg, 1987). In this view, the primary concern is the balance between network

---

[1]We consider task agnostic *future gradients*, referring to gradients of the model parameters with respect to unseen data points. These can be drawn from tasks that have already been partially learned or unseen tasks.

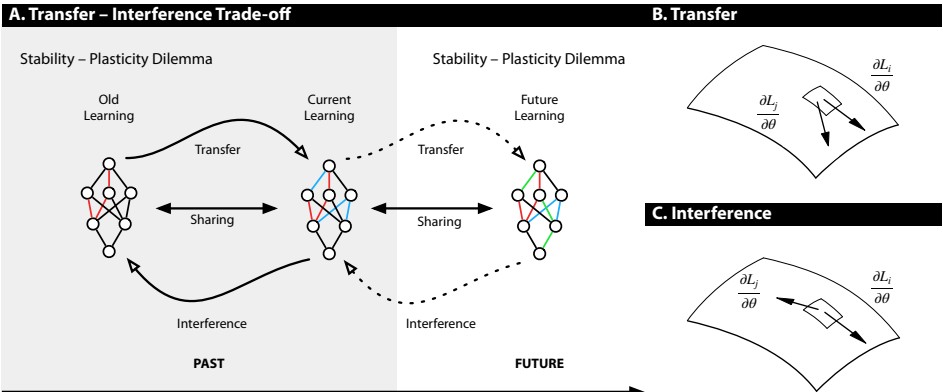

Figure 1: A) The stability-plasticity dilemma considers plasticity with respect to the current learning and how it degrades old learning. The transfer-interference trade-off considers the stability-plasticity dilemma and its dependence on weight sharing in both forward and backward directions. This symmetric view is crucial as solutions that purely focus on reducing the degree of weight-sharing are unlikely to produce transfer in the future. B) A depiction of transfer in weight space. C) A depiction of interference in weight space.

stability (to preserve past knowledge) and plasticity (to rapidly learn the current experience). For example, these techniques focus on balancing limited weight sharing with some mechanism to ensure fast learning (Li & Hoiem, 2016; Riemer et al., 2016a; Lopez-Paz & Ranzato, 2017; Rosenbaum et al., 2018; Lee et al., 2018; Serrà et al., 2018). In this paper, we extend this view by noting that for continual learning over an unbounded number of distributions, we need to consider weight sharing and the stability-plasticity trade-off in both the forward and backward directions in time (Figure 1A).

The transfer-interference trade-off proposed in this paper (section 2) presents a novel perspective on the goal of gradient alignment for the continual learning problem. This is right at the heart of the problem as these gradients are the update steps for SGD based optimizers during learning and there is a clear connection between gradients angles and managing the extent of weight sharing. The key difference in perspective with past conceptualizations of continual learning is that we are not just concerned with current transfer and interference with respect to past examples, but also with the dynamics of transfer and interference moving forward as we learn. Other approaches have certainly explored operational notions of transfer and interference in forward and backward directions (Lopez-Paz & Ranzato, 2017; Chaudhry et al., 2018), the link to weight sharing (French, 1991; Ajemian et al., 2013), and the idea of influencing gradient alignment for continual learning before (Lopez-Paz & Ranzato, 2017). However, in past work, ad hoc changes have been made to the dynamics of weight sharing based on current learning and past learning without formulating a consistent theory about the optimal weight sharing dynamics. This new view of the problem leads to a natural meta-learning (Schmidhuber, 1987) perspective on continual learning: we would like to learn to modify our learning to affect the dynamics of transfer and interference in a general sense. To the extent that our meta-learning into the future generalizes, this should make it easier for our model to perform continual learning in non-stationary settings. We achieve this by building off past work on experience replay (Murre, 1992; Lin, 1992; Robins, 1995) that has been a mainstay for solving non-stationary problems with neural networks. We propose a novel meta-experience replay (MER) algorithm that combines experience replay with optimization based meta-learning (section 3) as a first step towards modeling this perspective. Moreover, our experiments (sections 4, 5, and 6), confirm our theory. MER shows great promise across a variety of supervised continual learning and continual reinforcement learning settings. Critically, our approach is not reliant on any provided notion of tasks and in most of the settings we explore we must detect the concept of tasks without supervision. See Appendix B for a more detailed positioning with respect to related research.

## 2 THE TRANSFER-INTERFERENCE TRADE-OFF FOR CONTINUAL LEARNING

At an instant in time with parameters $\theta$ and loss $L$, we can define[2] operational measures of transfer and interference between two arbitrary distinct examples $(x_i, y_i)$ and $(x_j, y_j)$ while training with

---

[2]Throughout the paper we discuss ideas in terms of the supervised learning problem formulation. Extensions to the reinforcement learning formulation are straightforward. We provide more details in Appendix N.

SGD. *Transfer* occurs when:

$$\frac{\partial L(x_i, y_i)}{\partial \theta} \cdot \frac{\partial L(x_j, y_j)}{\partial \theta} > 0, \tag{1}$$

where $\cdot$ is the dot product operator. This implies that learning example $i$ will without repetition improve performance on example $j$ and vice versa (Figure 1B). *Interference* occurs when:

$$\frac{\partial L(x_i, y_i)}{\partial \theta} \cdot \frac{\partial L(x_j, y_j)}{\partial \theta} < 0. \tag{2}$$

Here, in contrast, learning example $i$ will lead to unlearning (i.e. forgetting) of example $j$ and vice versa (Figure 1C). [3] There is *weight sharing* between $i$ and $j$ when they are learned using an overlapping set of parameters. So, potential for transfer is maximized when weight sharing is maximized while potential for interference is minimized when weight sharing is minimized (Appendix C).

Past solutions for the stability-plasticity dilemma in continual learning operate in a simplified temporal context where learning is divided into two phases: all past experiences are lumped together as *old memories* and the data currently being learned qualifies as *new learning*. In this setting, the goal is to simply minimize the interference projecting backward in time, which is generally achieved by reducing the degree of weight sharing explicitly or implicitly. In Appendix D we explain how our baseline approaches (Kirkpatrick et al., 2017; Lopez-Paz & Ranzato, 2017) fit within this paradigm.

The important issue with this perspective, however, is that the system still has learning to do and what the future may bring is largely unknown. This makes it incumbent upon us to do nothing to potentially undermine the networks ability to effectively learn in an uncertain future. This consideration makes us extend the temporal horizon of the stability-plasticity problem forward, turning it, more generally, into a continual learning problem that we label as solving the *Transfer-Interference Trade-off* (Figure 1A). Specifically, it is important not only to reduce backward interference from our current point in time, but we must do so in a manner that does not limit our ability to learn in the future. This more general perspective acknowledges a subtlety in the problem: the issue of gradient alignment and thus weight sharing across examples arises both backward and forward in time. With this temporally symmetric perspective, the transfer-interference trade-off becomes clear. Here we propose a potential solution where we learn to learn in a way that promotes gradient alignment at each point in time. The weight sharing across examples that enables transfer to improve future performance must not disrupt performance on what has come previously. As such, our work adopts a meta-learning perspective on the continual learning problem. We would like to learn to learn each example in a way that generalizes to other examples from the overall distribution.

## 3 A SYSTEM FOR LEARNING TO LEARN WITHOUT FORGETTING

In typical offline supervised learning, we can express our optimization objective over the stationary distribution of $x, y$ pairs within the dataset $D$:

$$\theta = \arg\min_{\theta} \mathbb{E}_{(x,y) \sim D}[L(x, y)], \tag{3}$$

where $L$ is the loss function, which can be selected to fit the problem. If we would like to maximize transfer and minimize interference, we can imagine it would be useful to add an auxiliary loss to the objective to bias the learning process in that direction. Considering equations 1 and 2, one obviously beneficial choice would be to also directly consider the gradients with respect to the loss function evaluated at randomly chosen datapoints. If we could maximize the dot products between gradients at these different points, it would directly encourage the network to share parameters where gradient directions align and keep parameters separate where interference is caused by gradients in opposite directions. So, ideally we would like to optimize for the following objective [4]:

$$\theta = \arg\min_{\theta} \mathbb{E}_{[(x_i, y_i), (x_j, y_j)] \sim D}[L(x_i, y_i) + L(x_j, y_j) - \alpha \frac{\partial L(x_i, y_i)}{\partial \theta} \cdot \frac{\partial L(x_j, y_j)}{\partial \theta}], \tag{4}$$

---

[3] We borrow our terminology from operational measures of forward transfer and backward transfer in Lopez-Paz & Ranzato (2017), but adopt a temporally symmetric view of the phenomenon by dropping the specification of direction. Interference commonly refers to negative transfer in either direction in the literature.

[4] The inclusion of $L(x_j, y_j)$ is largely an arbitrary notation choice as the relative prioritization of the two types of terms can be absorbed in $\alpha$. We use this notation as it is most consistant with our implementation.

---

**Algorithm 1** Meta-Experience Replay (MER)

---

**procedure** TRAIN($D, \theta, \alpha, \beta, \gamma, s, k$)
    $M \leftarrow \{\}$
    **for** $t = 1, ..., T$ **do**
        **for** $(x, y)$ `in` $D_t$ **do**
            *// Draw batches from buffer:*
            $B_1, ..., B_s \leftarrow sample(x, y, s, k, M)$
            $\theta_0^A \leftarrow \theta$
            **for** $i = 1, ..., s$ **do**
                $\theta_{i,0}^W \leftarrow \theta$
                **for** $j = 1, ..., k$ **do**
                    $x_c, y_c \leftarrow B_i[j]$
                    $\theta_{i,j}^W \leftarrow SGD(x_c, y_c, \theta_{i,j-1}^W, \alpha)$
                **end for**
                *// Within batch Reptile meta-update:*
                $\theta \leftarrow \theta_{i,0}^W + \beta(\theta_{i,k}^W - \theta_{i,0}^W)$
                $\theta_i^A \leftarrow \theta$
            **end for**
            *// Across batch Reptile meta-update:*
            $\theta \leftarrow \theta_0^A + \gamma(\theta_s^A - \theta_0^A)$
            *// Reservoir sampling memory update:*
            $M \leftarrow M \cup \{(x, y)\}$ (algorithm 3)
        **end for**
    **end for**
    **return** $\theta, M$
**end procedure**

---

where $(x_i, y_i)$ and $(x_j, y_j)$ are randomly sampled unique data points. We will attempt to design a continual learning system that optimizes for this objective. However, there are multiple problems that must be addressed to implement this kind of learning process in practice. The first problem is that continual learning deals with learning over a non-stationary stream of data. We address this by implementing an experience replay module that augments online learning so that we can approximately optimize over the stationary distribution of all examples seen so far. Another practical problem is that the gradients of this loss depend on the second derivative of the loss function, which is expensive to compute. We address this by indirectly approximating the objective to a first order Taylor expansion using a meta-learning algorithm with minimal computational overhead.

### 3.1 EXPERIENCE REPLAY

**Learning objective:** The continual lifelong learning setting poses a challenge for the optimization of neural networks as examples come one by one in a non-stationary stream. Instead, we would like our network to optimize over the stationary distribution of all examples seen so far. Experience replay (Lin, 1992; Murre, 1992) is an old technique that remains a central component of deep learning systems attempting to learn in non-stationary settings, and we will adopt here conventions from recent work (Zhang & Sutton, 2017; Riemer et al., 2017b) leveraging this approach. The central feature of experience replay is keeping a memory of examples seen $M$ that is interleaved with the training of the current example with the goal of making training more stable. As a result, experience replay approximates the objective in equation 3 to the extent that $M$ approximates $D$:

$$\theta = arg \min_\theta \mathbb{E}_{(x,y) \sim M}[L(x, y)], \tag{5}$$

$M$ has a current size $M_{size}$ and maximum size $M_{max}$. In our work, we update the buffer with reservoir sampling (Appendix F). This ensures that at every time-step the probability that any of the $N$ examples seen has of being in the buffer is equal to $M_{size}/N$. The content of the buffer resembles a stationary distribution over all examples seen to the extent that the items stored captures the variation of past examples. Following the standard practice in offline learning, we train by randomly sampling a batch $B$ from the distribution captured by $M$.

**Prioritizing the current example:** the variant of experience replay we explore differs from offline learning in that the current example has a special role ensuring that it is always interleaved with the examples sampled from the replay buffer. This is because before we proceed to the next example, we want to make sure our algorithm has the ability to optimize for the current example (particularly if it is not added to the memory). Over $N$ examples seen, this still implies that we have trained with each example as the current example with probability per step of $1/N$. We provide algorithms further detailing how experience replay is used in this work in Appendix G (algorithms 4 and 5).

**Concerns about storing examples:** Obviously, it is not scalable to store every experience seen in memory. As such, in this work we focus on showing that we can achieve greater performance than baseline techniques when each approach is provided with only a small memory buffer.

### 3.2 COMBINING EXPERIENCE REPLAY WITH OPTIMIZATION BASED META-LEARNING

**First order meta-learning:** One of the most popular meta-learning algorithms to date is Model Agnostic Meta-Learning (MAML) (Finn et al., 2017). MAML is an optimization based meta-learning algorithm with nice properties such as the ability to approximate any learning algorithm and the

ability to generalize well to learning data outside of the previous distribution (Finn & Levine, 2017). One aspect of MAML that limits its scalability is the need to explicitly compute second derivatives. The authors proposed a variant called first-order MAML (FOMAML), which is defined by ignoring the second derivative terms to address this issue and surprisingly found that it achieved very similar performance. Recently, this phenomenon was explained by Nichol & Schulman (2018) who noted through Taylor expansion that the two algorithms were approximately optimizing for the same loss function. Nichol & Schulman (2018) also proposed an algorithm, Reptile, that efficiently optimizes for approximately the same objective while not requiring that the data be split into training and testing splits for each task learned as MAML does. Reptile is implemented by optimizing across $s$ batches of data sequentially with an SGD based optimizer and learning rate $\alpha$. After training on these batches, we take the initial parameters before training $\theta_0$ and update them to $\theta_0 \leftarrow \theta_0 + \beta * (\theta_k - \theta_0)$ where $\beta$ is the learning rate for the meta-learning update. The process repeats for each series of $s$ batches (algorithm 2). Shown in terms of gradients in Nichol & Schulman (2018), Reptile approximately optimizes for the following objective over a set of $s$ batches:

$$\theta = arg \min_{\theta} \mathbb{E}_{B_1,...,B_s \sim D}[2 \sum_{i=1}^{s} [L(B_i) - \sum_{j=1}^{i-1} \alpha \frac{\partial L(B_i)}{\partial \theta} \cdot \frac{\partial L(B_j)}{\partial \theta}]], \tag{6}$$

where $B_1,...,B_s$ are batches within $D$. This is similar to our motivation in equation 4 to the extent that gradients produced on these batches approximate samples from the stationary distribution.

**The MER learning objective:** In this work, we modify the Reptile algorithm to properly integrate it with an experience replay module, facilitating continual learning while maximizing transfer and minimizing interference. As we describe in more detail during the derivation in Appendix I, achieving the Reptile objective in an online setting where examples are provided sequentially is non-trivial and is in part only achievable because of our sampling strategies for both the buffer and batch. Following our remarks about experience replay from the prior section, this allows us to optimize for the following objective in a continual learning setting using our proposed MER algorithm:

$$\theta = arg \min_{\theta} \mathbb{E}_{[(x_{11},y_{11}),...,(x_{sk},y_{sk})] \sim M}[2 \sum_{i=1}^{s} \sum_{j=1}^{k} [L(x_{ij}, y_{ij}) - \sum_{q=1}^{i-1} \sum_{r=1}^{j-1} \alpha \frac{\partial L(x_{ij}, y_{ij})}{\partial \theta} \cdot \frac{\partial L(x_{qr}, y_{qr})}{\partial \theta}]]. \tag{7}$$

**The MER algorithm:** MER maintains an experience replay style memory $M$ with reservoir sampling and at each time step draws $s$ batches including $k-1$ random samples from the buffer to be trained alongside the current example. Each of the $k$ examples within each batch is treated as its own Reptile batch of size 1 with an inner loop Reptile meta-update after that batch is processed. We then apply the Reptile meta-update again in an outer loop across the $s$ batches. We provide further details for MER in algorithm 1. This procedure approximates the objective of equation 7 when $\beta = 1$. The *sample* function produces $s$ batches for updates. Each batch is created by first adding the current example and then interleaving $k-1$ random examples from $M$.

**Controlling the degree of regularization:** In light of our ideal objective in equation 4, we can see that using a SGD batch size of 1 has an advantage over larger batches because it allows for the second derivative information conveyed to the algorithm to be fine grained on the example level. Another reason to use sample level effective batches is that for a given number of samples drawn from the buffer, we maximize $s$ from equation 6. In equation 6, the typical offline learning loss has a weighting proportional to $s$ and the regularizer term to maximize transfer and minimize interference has a weighting proportional to $\alpha s(s-1)/2$. This implies that by maximizing the effective $s$ we can put more weight on the regularization term. We found that for a fixed number of examples drawn from $M$, we consistently performed better converting to a long list of individual samples than we did using proper batches as in Nichol & Schulman (2018) for few shot learning.

**Prioritizing current learning:** To ensure strong regularization, we would like our number of batches processed in a Reptile update to be large – enough that experience replay alone would start to overfit to $M$. As such, we also need to make sure we provide enough priority to learning the current example, particularly because we may not store it in $M$. To achieve this in algorithm 1, we sample $s$ separate batches from $M$ that are processed sequentially and each interleaved with the current example. In Appendix H we also outline two additional variants of MER with very similar properties in that they effectively approximate for the same objective. In one we choose one big batch of size $sk - s$ memories and $s$ copies of the current example (algorithm 6). In the other, we choose one memory batch of size $k-1$ with a special current item learning rate of $s\alpha$ (algorithm 7).

**Unique properties:** In the end, our approach amounts to a quite easy to implement and computationally efficient extension of SGD, which is applied to an experience replay buffer by leveraging the machinery of past work on optimization based meta-learning. However, the emergent regularization on learning is totally different than those previously considered. Past work on optimization based meta-learning has enabled fast learning on incoming data without considering past data. Meanwhile, past work on experience replay only focused on stabilizing learning by approximating stationary conditions without altering model parameters to change the dynamics of transfer and interference.

## 4 EVALUATION FOR SUPERVISED CONTINUAL LIFELONG LEARNING

To test the efficacy of MER we compare it to relevant baselines for continual learning of many supervised tasks from Lopez-Paz & Ranzato (2017) (see Appendix D for in-depth descriptions):

- **Online:** represents online learning performance of a model trained straightforwardly one example at a time on the incoming non-stationary training data by simply applying SGD.
- **Independent:** an independent predictor per task with less hidden units proportional to the number of tasks. When useful, it can be initialized by cloning the last predictor.
- **Task Input:** has the same architecture as Online, but with a dedicated input layer per task.
- **EWC:** Elastic Weight Consolidation (EWC) (Kirkpatrick et al., 2017) is an algorithm that modifies online learning where the loss is regularized to avoid catastrophic forgetting.
- **GEM:** Gradient Episodic Memory (GEM) (Lopez-Paz & Ranzato, 2017) is an approach for making efficient use of episodic storage by following gradients on incoming examples to the maximum extent while altering them so that they do not interfere with past memories. An independent adhoc analysis is performed to alter each incoming gradient. In contrast to MER, nothing generalizable is learned across examples about how to alter gradients.

We follow Lopez-Paz & Ranzato (2017) and consider final retained accuracy across all tasks after training sequentially on all tasks as our main metric for comparing approaches. Moving forward we will refer to this metric as *retained accuracy (RA)*. In order to reveal more characteristics of the learning behavior, we also report the *learning accuracy (LA)* which is the average accuracy for each task directly after it is learned. Additionally, we report the *backward transfer and interference (BTI)* as the average change in accuracy from when a task is learned to the end of training. A highly negative BTI reflects catastrophic forgetting. Forward transfer and interference (Lopez-Paz & Ranzato, 2017) is only applicable for one task we explore, so we provide details in Appendix K.

**Question 1** *How does MER perform on supervised continual learning benchmarks?*

To address this question we consider two continual learning benchmarks from Lopez-Paz & Ranzato (2017). **MNIST Permutations** is a variant of MNIST first proposed in Kirkpatrick et al. (2017) where each task is transformed by a fixed permutation of the MNIST pixels. As such, the input distribution of each task is unrelated. **MNIST Rotations** is another variant of MNIST proposed in Lopez-Paz & Ranzato (2017) where each task contains digits rotated by a fixed angle between 0 and 180 degrees. We follow the standard benchmark setting from Lopez-Paz & Ranzato (2017) using a modest memory buffer of size 5120 to learn 1000 sampled examples across each of 20 tasks. We provide detailed information about our architectures and hyperparameters in Appendix J.

In Table 1 we report results on these benchmarks in comparison to our baseline approaches. Clearly GEM outperforms our other baselines, but our approach adds significant value over GEM in terms of retained accuracy on both benchmarks. MER achieves this by striking a superior balance between transfer and interference with respect to the past and future data. MER displays the best adaption to incoming tasks, while also providing very strong retention of knowledge when learning future tasks. EWC and using a task specific input layer both also lead to gains over standard online learning in terms of retained accuracy. However, they are quite far below the performance of approaches that make usage of episodic storage. While EWC does not store examples, in storing the Fisher information for each task it accrues more incremental resources than the episodic storage approaches.

**Question 2** *How do the performance gains from MER vary as a function of the buffer size?*

To make progress towards the greater goals of lifelong learning, we would like our algorithm to make the most use of even a modest buffer. This is because in extremely large scale settings it is

| Model | MNIST Rotations | | | MNIST Permutations | | |
|---|---|---|---|---|---|---|
| | RA | LA | BTI | RA | LA | BTI |
| Online | 53.38 ± 1.53 | 58.82 ± 2.17 | -5.44 ± 1.70 | 55.42 ± 0.65 | 69.18 ± 0.99 | -13.76 ± 1.19 |
| Independent | 60.74 ± 4.55 | 60.74 ± 4.55 | - | 55.80 ± 4.79 | 55.80 ± 4.79 | - |
| Task Input | 79.98 ± 0.66 | 81.04 ± 0.34 | -1.06 ± 0.59 | 80.46 ± 0.80 | 81.20 ± 0.52 | -0.74 ± 1.10 |
| EWC | 57.96 ± 1.33 | 78.38 ± 0.72 | -20.42 ± 1.60 | 62.32 ± 1.34 | 75.64 ± 1.18 | -13.32 ± 2.24 |
| GEM | 87.58 ± 0.32 | 86.46 ± 0.46 | +1.12 ± 0.35 | 83.02 ± 0.23 | 80.46 ± 0.38 | **+2.56** ± 0.42 |
| MER | **89.56** ± 0.11 | **87.62** ± 0.16 | **+1.94** ± 0.18 | **85.50** ± 0.16 | **86.36** ± 0.21 | -0.86 ± 0.21 |

Table 1: Performance on continual lifelong learning 20 tasks benchmarks from (Lopez-Paz & Ranzato, 2017).

| Model | Buffer | MNIST Rotations | | | MNIST Permutations | | |
|---|---|---|---|---|---|---|---|
| | | RA | LA | BTI | RA | LA | BTI |
| GEM | 5120 | 87.58 ± 0.32 | 86.46 ± 0.46 | +1.12 ± 0.35 | 83.02 ± 0.23 | 80.46 ± 0.38 | **+2.56** ± 0.42 |
| | 500 | 74.88 ± 0.93 | 85.90 ± 0.53 | -11.02 ± 1.22 | 69.26 ± 0.66 | 80.28 ± 0.52 | -11.02 ± 0.71 |
| | 200 | 67.38 ± 1.75 | **85.40** ± 0.53 | -18.02 ± 1.99 | 55.42 ± 1.10 | 79.84 ± 1.01 | -24.42 ± 1.10 |
| MER | 5120 | **89.56** ± 0.11 | **87.62** ± 0.16 | **+1.94** ± 0.18 | **85.50** ± 0.16 | **86.36** ± 0.21 | -0.86 ± 0.21 |
| | 500 | **82.08** ± 0.31 | **85.66** ± 0.20 | **-3.58** ± 0.39 | **77.50** ± 0.46 | **83.90** ± 0.23 | **-6.40** ± 0.35 |
| | 200 | **77.42** ± 0.78 | 83.02 ± 0.23 | **-5.60** ± 0.70 | **73.46** ± 0.45 | **84.42** ± 0.20 | **-9.96** ± 0.45 |

Table 2: Performance varying the buffer size on continual learning benchmarks (Lopez-Paz & Ranzato, 2017).

unrealistic to assume a system can store a large percentage of previous examples in memory. As such, we would like to compare MER to GEM, which is known to perform well with an extremely small memory buffer (Lopez-Paz & Ranzato, 2017). We consider a buffer size of 500, that is over 10 times smaller than the standard setting on these benchmarks. Additionally, we also consider a buffer size of 200, matching the smallest setting explored in Lopez-Paz & Ranzato (2017). This setting corresponds to an average storage of 1 example for each combination of task and class. We report our results in Table 2. The benefits of MER seem to grow as the buffer becomes smaller. In the smallest setting, MER provides more than a 10% boost in retained accuracy on both benchmarks.

**Question 3** *How effective is MER at dealing with increasingly non-stationary settings?*

Another larger goal of lifelong learning is to enable continual learning with only relatively few examples per task. This setting is particularly difficult because we have less data to characterize each class to learn from and our distribution is increasingly non-stationary over a fixed amount of training. We would like to explore how various models perform in this kind of setting. To do this we consider two new benchmarks. **Many Permutations** is a variant of *MNIST Permutations* that has 5 times more tasks (100 total) and 5 times less training examples per task (200 each). Meanwhile we also explore the **Omniglot** (Lake et al., 2011) benchmark treating each of the 50 alphabets to be a task (see Appendix J for experimental details). Following multi-task learning conventions, 90% of the data is used for training and 10% is used for testing (Yang & Hospedales, 2017). Overall there are 1623 characters. We learn each character and task sequentially with a task specific output layer.

We report continual learning results using these new datasets in Table 3. The effect on *Many Permutations* of efficiently using episodic storage becomes even more pronounced when the setting becomes more non-stationary. GEM and MER both achieve nearly double the performance of EWC and online learning. We also see that increasingly non-stationary settings lead to a larger performance gain for MER over GEM. Gains are quite significant for *Many Permutations* and remarkable for *Omniglot*. *Omniglot* is even more non-stationary including slightly fewer examples per task and MER nearly quadruples the performance of baseline techniques. Considering the poor performance

| Model | Buffer | Many Permutations | | | Omniglot | | |
|---|---|---|---|---|---|---|---|
| | | RA | LA | BTI | RA | LA | BTI |
| Online | 0 | 32.62 ± 0.43 | 51.68 ± 0.65 | -19.06 ± 0.86 | 4.36 ± 0.37 | 5.38 ± 0.18 | -1.02 ± 0.33 |
| EWC | 0 | 33.46 ± 0.46 | 51.30 ± 0.81 | -17.84 ± 1.15 | 4.63 ± 0.14 | 9.43 ± 0.63 | -4.80 ± 0.68 |
| GEM | 5120 | 56.76 ± 0.29 | **59.66** ± 0.46 | -2.92 ± 0.52 | 18.03 ± 0.15 | 3.86 ± 0.09 | **+14.19** ± 0.19 |
| | 500 | 32.14 ± 0.50 | 55.66 ± 0.53 | -23.52 ± 0.87 | - | - | - |
| MER | 5120 | **62.52** ± 0.32 | 59.44 ± 0.23 | **+3.08** ± 0.31 | **75.23** ± 0.52 | **69.12** ± 0.83 | +6.11 ± 0.62 |
| | 500 | **47.40** ± 0.35 | **65.18** ± 0.20 | **-17.78** ± 0.39 | **32.05** ± 0.69 | **28.78** ± 0.91 | +3.27 ± 1.04 |

Table 3: Performance on many task non-stationary continual lifelong learning benchmarks.

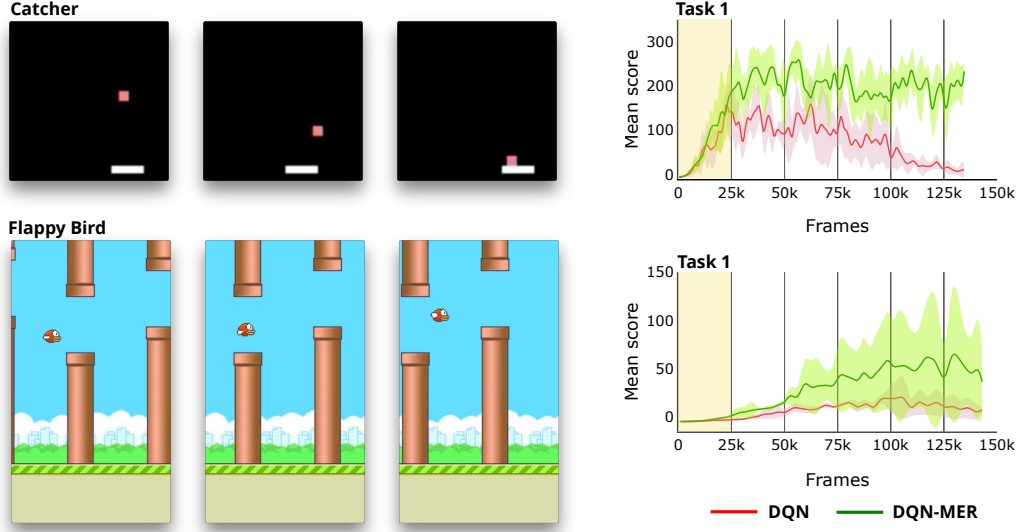

Figure 2: *Left*: a sequence of frames from Catcher and Flappy Bird respectively. The goal in Catcher is to capture the falling pellet by moving the racket on the bottom of the screen. In Flappy Bird, the goal is to navigate the bird through as many pipes as possible by making it go up or letting it fall. *Right*: average score in Catcher (above) and Flappy Bird (below) for evaluation on the first task which has slower falling pellets and a larger pipe gap.

of online learning and EWC it is natural to question whether or not examples were learned in the first place. We experiment with using as many as 100 gradient descent steps per incoming example to ensure each example is learned when first seen. However, due to the extremely non-stationary setting no run of any variant we tried surpassed 5.5% retained accuracy. GEM also has major deficits for learning on Omniglot that are resolved by MER which achieves far better performance when it comes to quickly learning the current task. GEM maintains a buffer using a recent item based sampling strategy and thus can not deal with non-stationarity within the task nearly as well as reservoir sampling. Additionally, we found that the optimization based on the buffer was significantly less effective and less reliable as the quadratic program fails for many hyperparameter values that lead to non-positive definite matrices. Unfortunately, we could not get GEM to consistently converge on Omniglot for a memory size of 500 (significantly less than the number of classes), meanwhile MER handles it well. In fact, MER greatly outperforms GEM with an order of magnitude smaller buffer.

We provide additional details about our experiments on Omniglot in Figure 3. We plot retained training accuracy, retained testing accuracy, and computation time for the entire training period using one CPU. We find that MER strikes the best balance of computational efficiency and performance even when using algorithm 1 for MER which performs more computation than algorithm 7. The computation involved in the GEM update does not scale well to large CNN models like those that are popular for Omniglot. MER is far better able to fit the training

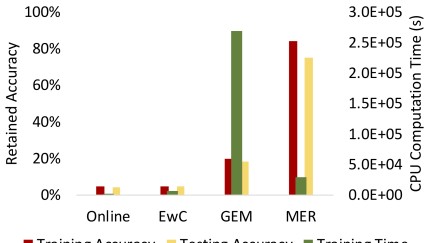

Figure 3: Further details on Omniglot performance characteristics for each model.

data than our baseline models while maintaining a computational efficiency closer to online update methods like EWC than GEM.

## 5 EVALUATION FOR CONTINUAL REINFORCEMENT LEARNING

**Question 4** *Can MER improve a DQN with ER in continual reinforcement learning settings?*

We considered the evaluation of MER in a continual reinforcement learning setting where the environment is highly non-stationary. In order to produce these non-stationary environments in a

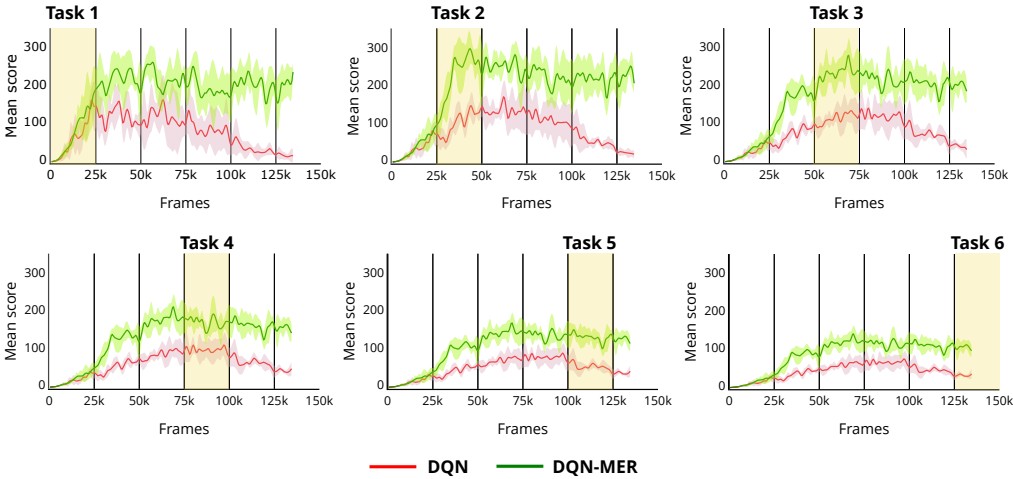

Figure 4: Continual learning performance for a non-stationary version of Catcher. Graphs show averaged values over ten validation episodes across five different seeds. Vertical grid lines on the x-axis indicate a task switch.

controlled way suitable for our experimental purposes, we utilized arcade games provided by Tasfi (2016). Specifically, we used Catcher and Flappy Bird, two simple but interesting enough environments (see Appendix N.1 for details). For the purposes of our explanation, we will call each set of fixed game-dependent parameters a task[5]. The multi-task setting is then built by introducing changes in these parameters, resulting in non-stationarity across tasks. Each agent is once again evaluated based on its performance over time on all tasks. Our model uses a standard DQN model, developed for Atari (Mnih et al., 2015). See Appendix N.2 for implementation details.

In Catcher, we then obtain different tasks by incrementally increasing the pellet velocity a total of 5 times during training. In Flappy Bird, the different tasks are obtained by incrementally reducing the separation between upper and lower pipes a total of 5 times during training. In Figure 4, we show the performance in Catcher when trained sequentially on 6 different tasks for 25k frames each to a maximum of 150k frames, evaluated at each point in time in all 6 tasks. Under these non-stationary conditions, a DQN using MER performs consistently better than the standard DQN with an experience replay buffer (see Appendix N.4 for further comments and ablation results). If we take as inspiration how humans perform, in the last stages of training we hope that a player that obtains good results in later tasks will also obtain good results in the first tasks, as the first tasks are subsumed in the latter ones. For example, in Catcher, the pellet moves faster in later tasks, and thus we expect to be able to do well on the first task. However, DQN forgets significantly how to get slowly moving pellets. In contrast, DQN-MER exhibits minimal or no forgetting after training on the rest of the tasks. This behavior is intuitive because we would expect transfer to happen naturally in this setting. We see similar behavior for Flappy Bird. DQN-MER becomes a Platinum player on the first task when it is learning the third task. This is a more difficult environment in which the pipe gap is noticeably smaller (see Appendix N.4). DQN-MER exhibits the kind of learning patterns expected from humans for these games, while a standard DQN struggles to generalize as the game changes and to retain knowledge over time.

## 6    FURTHER ANALYSIS OF THE APPROACH

In this section we would like to dive deeper into how MER works. To do so we run additional detailed experiments across our three MNIST based continual learning benchmarks.

**Question 5** *Does MER lead to a shift in the distribution of gradient dot products?*

We would like to directly verify that MER achieves our motivation in equation 7 and results in significant changes in the distribution of gradient dot products between new incoming examples and past examples over the course of learning when compared to experience replay (ER) from algorithm

---

[5]Agents are not provided task information, forcing them to identify changes in game play on their own.

| Model | MNIST Permutations | MNIST Rotations | Many Permutations |
|-------|--------------------|-----------------|-------------------|
| ER | -0.569 $_{\pm\,0.077}$ | -1.652 $_{\pm\,0.082}$ | -1.280 $_{\pm\,0.078}$ |
| MER | +0.042 $_{\pm\,0.017}$ | +0.017 $_{\pm\,0.007}$ | +0.131 $_{\pm\,0.027}$ |

Table 4: Analysis of the mean dot product across the period of learning between gradients on incoming examples and gradients on randomly sampled past examples across 5 runs on MNIST based benchmarks.

5. For these experiments, we maintain a history of all examples seen that is totally separate from our notion of memory buffers that only include a partial history of examples. Every time we receive a new example we use the current model to extract a gradient direction and we also randomly sample five examples from the previous history. We save the dot products of the incoming example gradient with these five past example gradients and consider the mean of the distribution of dot products seen over the course of learning for each model. We run this experiment on the best hyperparamater setting for both ER and MER from algorithm 6 with one batch per example for fair comparison. Each model is evaluated five times over the course of learning. We report mean and standard deviations of the mean gradient dot product across runs in Table 4. We can thus verify that a very significant and reproducible difference in the mean gradient encountered is seen for MER in comparison to ER alone. This difference alters the learning process making incoming examples on average result in slight transfer rather than significant interference. This analysis confirms the desired effect of the objective function in equation 7. For these tasks there are enough similarities that our meta-learning generalizes very well into the future. We should also expect it to perform well during drastic domain shifts like other meta-learning algorithms driven by SGD alone (Finn & Levine, 2017).

**Question 6** *What components of MER are most important?*

We would like to further analyze our MER model to understand what components add the most value and when. We want to understand how powerful our proposed variants of ER are on their own and how much is added by adding meta-learning to ER. In Appendix L we provide detailed results considering ablated baselines for our experiments on the MNIST lifelong learning benchmarks. [6] Our versions of ER consistently provide gains over GEM on their own, but the techniques perform very comparably when we also maintain GEM's buffer with reservoir sampling or use ER with a GEM style buffer. Additionally, we see that adding meta-learning to ER consistently results in performance gains. In fact, meta-learning appears to provide increasing value for smaller buffers. In Appendix M, we provide further validation that our results are reproducible across runs and seeds.

We would also like to compare the variants of MER proposed in algorithms 1, 6, and 7. Conceptually algorithms 1 and 7 represent different mechanisms of increasing the importance of the current example in algorithm 6. We find that all variants of MER result in significant improvements on ER. Meanwhile, the variants that increase the importance of the current example see a further improvement in performance, performing quite comparably to each other. Overall, in our MNIST experiments algorithm 7 displays the best tradeoff of computational efficiency and performance. Finally, we conducted experiments demonstrating that adaptive optimizers like Adam and RMSProp can not account for the gap between ER and MER. Particularly for smaller buffer sizes, these approaches overfit more on the buffer and actually hurt generalization in comparison to SGD.

## 7 CONCLUSION

In this paper we have cast a new perspective on the problem of continual learning in terms of a fundamental trade-off between transfer and interference. Exploiting this perspective, we have in turn developed a new algorithm Meta-Experience Replay (MER) that is well suited for application to general purpose continual learning problems. We have demonstrated that MER regularizes the objective of experience replay so that gradients on incoming examples are more likely to have transfer and less likely to have interference with respect to past examples. The result is a general purpose solution to continual learning problems that outperforms strong baselines for both supervised continual learning benchmarks and continual learning in non-stationary reinforcement learning environments. Techniques for continual learning have been largely driven by different conceptualizations of the fundamental problem encountered by neural networks. We hope that the transfer-interference trade-off can be a useful problem view for future work to exploit with MER as a first successful example.

---

[6]Code available at `https://github.com/mattriemer/mer`.

ACKNOWLEDGMENTS

We would like to thank Pouya Bashivan, Christopher Potts, Dan Jurafsky, and Joshua Greene for their input and support of this work. Additionally, we would like to thank Arslan Chaudhry and Marc'Aurelio Ranzato for their helpful comments and discussions. We also thank the three anonymous reviewers for their valuable suggestions. This research was supported by the MIT-IBM Watson AI Lab, and is based in part upon work supported by the Stanford Data Science Initiative and by the NSF under Grant No. BCS-1456077 and the NSF Award IIS-1514268.

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

## A    CONTINUAL LEARNING PROBLEM FORMULATION

In the classical *offline supervised learning* setting, a learning agent is given a fixed training data set $D = \{(\boldsymbol{x}_i, y_i)\}_{i=1}^n$ of $n$ samples, each containing an input feature vector $\boldsymbol{x}_i \in \mathcal{X}$ associated with the corresponding output (target, or label) $y_i \in \mathcal{Y}$; a common assumption is that the training samples are i.i.d. samples drawn from the same unknown joint probability distribution $P(\boldsymbol{x}, y)$. The learning task is often formulated as a *function approximation* problem, i.e. finding a function, or model, $f_{\boldsymbol{\theta}}(\boldsymbol{x}) : \mathcal{X} \rightarrow \mathcal{Y}$ from a given class of models (e.g., neural networks, decision trees, linear functions, etc.) where $\boldsymbol{\theta}$ are the parameters estimated from data. Given a loss function $L(f_{\boldsymbol{\theta}}(\boldsymbol{x}), y)$, the parameter estimation is formulated as an empirical risk minimization problem: $\min_{\boldsymbol{\theta}} \frac{1}{|D|} \sum_{(\mathbf{x_i}, \mathbf{y_i}) \sim \mathbf{D}} L(f_{\boldsymbol{\theta}}(\boldsymbol{x}), y)$.

On the contrary, the *online learning* setting does not assume a fixed training dataset but rather a stream of data samples, where unlabeled feature vectors arrive one at a time, or in small mini-batches, and the learner must assign labels to those inputs, receive the correct labels, and update the model accordingly, in iterative fashion. While classical online learning assumes i.i.d. samples, *continual* or *lifelong learning* does not make such an assumption, and requires a learning agent to handle non-stationary data streams.

In this work, we define *continual learning* as online learning from a non-stationary input data stream, with a specific type of non-stationarity as defined below. Namely, we follow a commonly used setting to define non-stationary conditions for continual learning, dubbed *locally i.i.d* by Lopez-Paz & Ranzato (2017), where the agent learns over a sequence of separate stationary distributions one after another. We call the individual stationary distributions *tasks*, where each task $t_k$ is an online supervised learning problem associated with its own data probability distribution $P_k(\boldsymbol{x}, y)$. Namely, we are given a (potentially infinite) sequence

$$(\boldsymbol{x}_1, y_1, t_1), ..., (\boldsymbol{x}_i, y_i, t_i), ..., (\boldsymbol{x}_{i+j}, y_{i+j}, t_{i+j})$$

While many continual learning methods assume the task descriptors $t_k$ are available to a learner, we are interested in developing approaches which do not have to rely on such information and can learn continuously without explicit announcement of the task change. Borrowing terminology from Chaudhry et al. (2018), we explore the *single-headed* setting in most of our experiments, which keeps learning a common function $f_\theta$ across changing tasks. In contrast, *multi-headed* learning, which we consider for our Omniglot experiments, involves a separate final classification layer for each task. This makes more sense in case of Omniglot dataset, where the number of classes for each task varies considerably from task to task. We should also note that for Omniglot we consider a setting that is locally i.i.d. at the class level rather than the task level.

## B    RELATION TO PAST WORK

With regard to the continual learning setting specifically, other recent work has explored similar operational measures of transfer and interference. For example, the notions of *Forward Transfer* and *Backward Transfer* were explored in Lopez-Paz & Ranzato (2017). However, the approach of that work, GEM, was primarily concerned with solving the classic stability-plasticity dilemma (Carpenter & Grossberg, 1987) at a specific instance of time. Adjustments to gradients on the current data are made in an ad hoc manner solving a quadratic program separately for each example.

In our work we try to learn a generalizable theory about weight sharing that can learn to influence the distribution of gradients not just in the past and present, but in the future as well. Additionally, in Chaudhry et al. (2018) similar ideas were explored with operational measures of *intransigence* (the inability to learn new data) and *forgetting* (the loss of previous performance). These measures are also intimately related to the stability-plasticity dilemma as *intransigence* is high when plasticity is low and *forgetting* is high when stability is low. The major distinction in the transfer-interference trade-off proposed in this work is that we aim to learn the optimal weight sharing scheme to optimize for the stability-plasticity dilemma with the hope that our learning about weight sharing will improve the stability and efficacy of learning on unseen data as well.

With regard to the problem of weight-sharing in neural networks more generally, a host of different strategies have been proposed in the past to deal with the problems of catastrophic forgetting and/or the stability-plasticity dilemma (for review, see French (1999)). For example, one strategy for alleviating catastrophic forgetting is to make distributed representations less distributed – or semi-distributed (French, 1991) – for the case of past learning. Activation sharpening as introduced by French (1991) is a prominent example. A second strategy known as dual network models (McClelland et al., 1995; Ans & Rousset, 1997) is based on the neurobiological finding that both hippocampal and cortical circuits contributed differentially to memory. The cortical circuits are highly distributed with overlapping representations suitable for task generalization, while the more sparse hippocampal representations tend to be non-overlapping. The existence of dual circuits provides an extra degree of freedom for balancing the dual constraints of stability and plasticity. In a similar spirit, models have been proposed that have two classes of weights operating on two different timescales (Hinton & Plaut, 1987). A third strategy also motivated by neurobiological considerations is the use of latent synaptic dynamics (Fusi et al., 2005; Lahiri & Ganguli, 2013). Here the basic idea is that synaptic strength is determined by a multiple of variables, including latent ones not easily observed, operating at different timescales such that their net effect is to provide the system with additional degrees-of-freedom to store past experience without interfering with current learning. A fourth strategy is the use of feedback mechanisms to stabilize representations (Carpenter & Grossberg, 1987; Murre, 1992). In this class of models, a previously experienced memory will trigger top down feedback that prevents plasticity, while novel stimuli that experience no such feedback trigger plasticity. All of these approaches have their own strengths and weaknesses with respect to the stability-plasticity dilemma and, by extension, the transfer-interference trade-off we propose.

Another relevant work is the POWERPLAY framework (Schmidhuber, 2004; 2013) which is a method for asymptotically optimal curriculum learning that by definition cannot forget previously learned skills. POWERPLAY also uses environment-independent replay of behavioral traces to avoid forgetting previous skills. However, POWERPLAY is orthogonal to our work as we consider a different setting where the agent cannot directly control the new tasks that will be encountered in the environment and thus must instead learn to adapt and react to non-stationarity conditions.

In contrast to past work on meta-learning for few shot learning (Santoro et al., 2016; Vinyals et al., 2016; Ravi & Larochelle, 2016; Finn et al., 2017) and reinforcement learning across successive tasks (Al-Shedivat et al., 2018), we are not only trying to improve the speed of learning on new data, but also trying to do it in a way that preserves knowledge of past data and generalizes to future data. While past work has considered learning to influence gradient angles, so that there is more alignment and thus faster learning within a task, we focus on a setting where we would like to influence gradient angles from all tasks at all points in time.

As our model aims to influence the dynamics of weight sharing, it bears conceptual similarity to mixtures of experts (Jacobs et al., 1991) style models for lifelong and multi-task learning (Misra et al., 2016; Riemer et al., 2016b; Aljundi et al., 2017; Fernando et al., 2017; Shazeer et al., 2017; Rosenbaum et al., 2018). MER implicitly affects the dynamics of weight sharing, but it is possible that combining it with mixtures of experts models could further amplify the ability for the model to control these dynamics. This is potentially an interesting avenue for future work.

The options framework has also been considered as a solution to a similar continual RL setting to the one we explore (Mankowitz et al., 2018). Options formalize the notion of temporally abstraction actions in RL. Interestingly, generic architectures designed for shallow (Bacon et al., 2017) or deep (Riemer et al., 2018) hierarchies of options in essence learn very complex patterns of weight sharing over time. The option hierarchies constitute an explicit mechanism of controlling the extent of weight sharing for continual learning, allowing for orthogonalization of weights relating to different

skills. In contrast, our work explores a method of implicitly optimizing weight sharing for continual learning that improves the efficacy of experience replay. MER should be simple to implement in concert with options based methods and combining the two is an interesting direction for future work.

## C  THE CONNECTION BETWEEN WEIGHT SHARING AND THE TRANSFER-INTERFERENCE TRADE-OFF

In this section we would like to generalize our interpretation of a large set of different weight sharing schemes including (Riemer et al., 2015; Bengio et al., 2015; Rosenbaum et al., 2018; Serrà et al., 2018) and how the concept of weight sharing impacts the dynamics of transfer (equation 1) and interference (equation 2). We will assume that we have a total parameter space $\theta$ that can be used by our network at any point in time. However, it is not a requirement that all parameters are actually used at all points in time. So, we can consider two specific instances in time. One where we receive data point $(x_1, y_1)$ and leverage parameters $\theta_1$. Then, at the other instance in time, we receive data point $(x_2, y_2)$ and leverage parameters $\theta_2$. $\theta_1$ and $\theta_2$ are both subsets of $\theta$ and critically the overlap between these subsets influences the possible extent of transfer and interference when training on either data point.

First let us consider two extremes. In the first extreme imagine $\theta_1$ and $\theta_2$ are entirely non-overlapping. As such $\frac{\partial L(x_1, y_1)}{\partial \theta} \cdot \frac{\partial L(x_2, y_2)}{\partial \theta} = 0$. On the positive side, this means that our solution has no potential for interference between the examples. On the other hand, there is no potential for transfer either. On the other extreme, we can imagine that $\theta_1 = \theta_2$. In this case, the potential for both transfer and interference is maximized as gradients with respect to every parameter have the possibility of a non-zero dot product with each other.

From this discussion it is clear that both the extreme of full weight sharing and the extreme of no weight sharing have value depending on the relationship between data points. What we would really like for continual learning is to have a system that learns when to share weights and when not to on its own. To the extent that our learning about weight sharing generalizes, this should allow us to find an optimal solution to the transfer-interference trade-off.

## D  FURTHER DESCRIPTIONS AND COMPARISONS WITH BASELINE ALGORITHMS

**Independent:** originally reported in (Lopez-Paz & Ranzato, 2017) is the performance of an independent predictor per task which has the same architecture but with less hidden units proportional to the number of tasks. The independent predictor can be initialized randomly or clone the last trained predictor depending on what leads to better performance.

**EWC:** Elastic Weight Consolidation (EWC) (Kirkpatrick et al., 2017) is an algorithm that modifies online learning where the loss is regularized to avoid catastrophic forgetting by considering the importance of parameters in the model as measured by their fisher information. EWC follows the catastrophic forgetting view of the continual learning problem by promoting less sharing of parameters for *new learning* that were deemed to be important for performance on *old memories*. We utilize the code provided by Lopez-Paz & Ranzato (2017) in our experiments. The only difference in our setting is that we provide the model one example at a time to test true continual learning rather than providing a batch of 10 examples at a time.

**GEM:** Gradient Episodic Memory (GEM) (Lopez-Paz & Ranzato, 2017) is an algorithm meant to enhance the effectiveness of episodic storage based continual learning techniques by allowing the model to adapt to incoming examples using SGD as long as the gradients do not interfere with examples from each task stored in a memory buffer. If gradients interfere leading to a decrease in the performance of a past task, a quadratic program is used to solve for the closest gradient to the original that does not have negative gradient dot products with the aggregate memories from any previous tasks. GEM is known to achieve superior performance in comparison to other recently proposed techniques that use episodic storage like Rebuffi et al. (2017), making superior use of small memory buffer sizes. GEM follows similar motivation to our approach in that it also considers the intelligent use of gradient dot product information to improve the use case of supervised continual

learning. As a result, it is a very strong and interesting baseline to compare with our approach. We modify the original code and benchmarks provided by Lopez-Paz & Ranzato (2017). Once again the only difference in our setting is that we provide the model one example at a time to test true continual learning rather than providing a batch of 10 examples at a time.

We can consider the GEM algorithm as tailored to the stability-plasticity dilemma conceptualization of continual learning in that it looks to preserve performance on past tasks while allowing for maximal plasticity to the new task. To achieve this, GEM solves a quadratic program to find an approximate gradient $g_{new}$ that closely matches $\frac{\partial L(x_{new}, y_{new})}{\partial \theta}$ while ensuring that the following constraint holds:

$$g_{new} \cdot \frac{\partial L(x_{old}, y_{old})}{\partial \theta} > 0. \tag{8}$$

## E    REPTILE ALGORITHM

We detail the standard Reptile algorithm from (Nichol & Schulman, 2018) in algorithm 2. The $sample$ function randomly samples $s$ batches of size $k$ from dataset $D$. The $SGD$ function applies min-batch stochastic gradient descent over a batch of data given a set of current parameters and learning rate.

---
**Algorithm 2** Reptile for Stationary Data
---
**procedure** TRAIN($D, \theta, \alpha, \beta, s, k$)
    **while** not done **do**
        // Draw batches from data:
        $B_1, ..., B_s \leftarrow sample(D, s, k)$
        $\theta_0 \leftarrow \theta$
        **for** $i = 1, ..., s$ **do**
            $\theta_i \leftarrow SGD(B_i, \theta_{i-1}, \alpha)$
        **end for**
        // Reptile meta-update:
        $\theta \leftarrow \theta_0 + \beta(\theta_s - \theta_0)$
    **end while**
    **return** $\theta$
**end procedure**

---

## F    DETAILS ON RESERVOIR SAMPLING

Throughout this paper we refer to updates to our memory $M$ as $M \leftarrow M \cup \{(x, y)\}$. We would like to now provide details on how we update our memory buffer using reservoir sampling as outlined in Vitter (1985) (algorithm 3). Reservoir sampling solves the problem of keeping some limited number $M$ of $N$ total items seen before with equal probability $\frac{M}{N}$ when you don't know what number $N$ will be in advance. The $randomInteger$ function randomly draws an integer inclusively between the provided minimum and maximum values.

---

**Algorithm 3** Reservoir Sampling with Algorithm R

---

**procedure** RESERVOIR($M, N, x, y$)
    **if** $M > N$ **then**
        $M[N] \leftarrow (x, y)$
    **else**
        $j = randomInteger(min = 0, max = N)$

        **if** $j < M$ **then**
            $M[j] \leftarrow (x, y)$
        **end if**
    **end if**
    **return** $M$
**end procedure**

---

# G   EXPERIENCE REPLAY ALGORITHMS

We detail the our variant of the experience replay in algorithm 4. This procedure closely follows recent enhancements discussed in Zhang & Sutton (2017); Riemer et al. (2017b;a) The $sample$ function randomly samples $k - 1$ examples from the memory buffer $M$ and interleaves them with the current example to form a single size $k$ batch. The $SGD$ function applies mini-batch stochastic gradient descent over a batch of data given a set of current parameters and learning rate.

---

**Algorithm 4** Experience Replay (ER) with Reservoir Sampling

---

**procedure** TRAIN($D, \theta, \alpha, k$)
    $M \leftarrow \{\}$
    **for** $t = 1, ..., T$ **do**
        **for** $(x, y)$ in $D_t$ **do**
            // Draw batch from buffer:
            $B \leftarrow sample(x, y, k, M)$
            // Update parameters with mini-batch SGD:
            $\theta \leftarrow SGD(B, \theta, \alpha)$
            // Reservoir sampling memory update:
            $M \leftarrow M \cup \{(x, y)\}$ (algorithm 3)
        **end for**
    **end for**
    **return** $\theta, M$
**end procedure**

---

Unfortunately, it is not straightforward to implement algorithm 4 in all circumstances. In particular, it depends whether the neural network architecture is single headed (sharing an output layer and output space among all tasks) or multi-headed (where each task gets its own unique output space). In multi-headed settings, it is common to consider the tasks in separate batches and to equally weight the sampled tasks during each update. This results in training the parameters evenly for each task and is particularly important so we pay equal attention to each set of task specific parameters. We detail an approach that separates tasks into sub-batches for a balanced update in algorithm 5. Here $L$ is the loss given a set of parameters over a batch of data and $SGD$ applies a mini-batch gradient descent update rule over a loss given a set of parameters and learning rate.

---

**Algorithm 5** Experience Replay (ER) with Tasks

---

   **procedure** TRAIN($D, \theta, \alpha, k$)
      $M \leftarrow \{\}$
      **for** $t = 1, ..., T$ **do**
         **for** $(x, y)$ in $D_t$ **do**
            // Draw batch from buffer:
            $B \leftarrow sample(x, y, k, M)$
            // Compute balanced loss across tasks
            loss = 0.0
            **for** $task$ in $B$ **do**
               $loss = loss + L(B[task], \theta)$
            **end for**
            // Update parameters with mini-batch SGD:
            $\theta \leftarrow SGD(loss, \theta, \alpha)$
            // Reservoir sampling memory update:
            $M \leftarrow M \cup \{(x, y)\}$ (algorithm 3)
         **end for**
      **end for**
      **return** $\theta, M$
   **end procedure**

---

Our experiments demonstrate that both variants of experience replay are very effective for continual learning. Meanwhile, each performs significantly better than the other on some datasets and settings.

# H  THE VARIANTS OF MER

We detail two additional variants of MER (algorithm 1) in algorithms 6 and 7. The $sample$ function takes on a slightly different meaning in each variant of the algorithm. In algorithm 1 $sample$ is used to produce $s$ batches consisting of $k - 1$ random examples from the memory buffer and the current example. In algorithm 6 $sample$ is used to produce one batch consisting of $sk - s$ examples from the memory buffer and $s$ copies of the current example. In algorithm 7 $sample$ is used to produce one batch consisting of $k - 1$ examples from the memory buffer. In algorithm 6, $sample$ places the current example at the end of the batch. Meanwhile, in algorithm 7, $sample$ places the current example in a random location within the batch. In contrast, the $SGD$ function carries a common meaning across algorithms, applying stochastic gradient descent over a particular input and output given a set of current parameters and learning rate.

---

**Algorithm 6** Meta-Experience Replay (MER) - One Big Batch

---

**procedure** TRAIN($D, \theta, \alpha, \gamma, sk$)
    $M \leftarrow \{\}$
    **for** $t = 1, ..., T$ **do**
        **for** $(x, y)$ in $D_t$ **do**
            // Draw batch from buffer:
            $B \leftarrow sample(x, y, s, k, M)$
            $\theta_0 \leftarrow \theta$
            **for** $i = 1, ..., sk$ **do**
                $x_c, y_c \leftarrow B_i[j]$
                $\theta_i \leftarrow SGD(x_c, y_c, \theta_{i-1}, \alpha)$
            **end for**
            // Reptile meta-update:
            $\theta \leftarrow \theta_0 + \gamma(\theta_{sk} - \theta_0)$
            // Reservoir sampling memory update:
            $M \leftarrow M \cup \{(x, y)\}$ (algorithm 3)
        **end for**
    **end for**
    **return** $\theta, M$
**end procedure**

---

---

**Algorithm 7** Meta-Experience Replay (MER) - Current Example Learning Rate

---

**procedure** TRAIN($D, \theta, \alpha, \gamma, s, k$)
    $M \leftarrow \{\}$
    **for** $t = 1, ..., T$ **do**
        **for** $(x, y)$ in $D_t$ **do**
            // Draw batch from buffer:
            $B, index \leftarrow sample(k - 1, M)$
            $\theta_0 \leftarrow \theta$
            // SGD on individual samples from batch:
            **for** $i = 1, ..., k - 1$ **do**
                $x_c, y_c \leftarrow B_i[j]$
                **if** $j = index$
                    // High learning rate SGD on current example:
                    $\theta_k \leftarrow SGD(x, y, \theta_{k-1}, s\alpha)$
                **else**
                    $\theta_i \leftarrow SGD(x_c, y_c, \theta_{i-1}, \alpha)$
            **end for**
            // Reptile meta-update:
            $\theta \leftarrow \theta_0 + \gamma(\theta_k - \theta_0)$
            // Reservoir sampling memory update:
            $M \leftarrow M \cup \{(x, y)\}$ (algorithm 3)
        **end for**
    **end for**
    **return** $\theta, M$
**end procedure**

---

## I   DERIVING THE EFFECTIVE OBJECTIVE OF MER

We would like to derive what objective Meta-Experience Replay (algorithm 1) approximates and show that it is approximately the same objective from algorithms 6 and 7. We follow conventions from Nichol & Schulman (2018) and first demonstrate what happens to the effective gradients computed by the algorithm in the most trivial case. As in Nichol & Schulman (2018), this allows us to extrapolate an effective gradient that is a function of the number of steps taken. We can then

consider the effective loss function that results in this gradient. Before we begin, let us define the following terms from Nichol & Schulman (2018):

$$g_i = \frac{\partial L(\theta_i)}{\partial \theta_i} \text{ (gradient obtained during SGD)} \tag{9}$$

$$\theta_{i+1} = \theta_i - \alpha g_i \text{ (sequence of parameter vectors)} \tag{10}$$

$$\bar{g}_i = \frac{\partial L(\theta_i)}{\partial \theta_0} \text{ (gradient at initial point)} \tag{11}$$

$$g_i^j = \frac{\partial L(\theta_i)}{\partial \theta_j} \text{ (gradient evaluated at point i with respect to parameters j)} \tag{12}$$

$$\bar{H}_i = \frac{\partial^2 L(\theta_i)}{\partial \theta_0^2} \text{ (Hessian at initial point)} \tag{13}$$

$$H_i^j = \frac{\partial^2 L(\theta_i)}{\partial \theta_j^2} \text{ (Hessian evaluated at point i with respect to parameters j)} \tag{14}$$

In Nichol & Schulman (2018) they consider the effective gradient across one loop of reptile with size $k = 2$. As we have both an outer loop of Reptile applied across batches and an inner loop applied within the batch to consider, we start with a setting where the number of batches $s = 2$ and the number of examples per batch $k = 2$. Let's recall from the original paper that the gradients of Reptile with $k = 2$ was:

$$g_{Reptile,k=2,s=1} = g_0 + g_1 = \bar{g}_0 + \bar{g}_1 - \alpha \bar{H}_1 \bar{g}_0 + O(\alpha^2) \tag{15}$$

So, we can also consider the gradients of Reptile if we had 4 examples in one big batch (algorithm 6) as opposed to 2 batches of 2 examples:

$$g_{Reptile,k=4,s=1} = g_0 + g_1 + g_2 + g_3$$
$$= \bar{g}_0 + \bar{g}_1 + \bar{g}_2 + \bar{g}_3 - \alpha \bar{H}_1 \bar{g}_0 - \alpha \bar{H}_2 \bar{g}_0 - \alpha \bar{H}_2 \bar{g}_1 - \alpha \bar{H}_3 \bar{g}_0 - \alpha \bar{H}_3 \bar{g}_1 - \alpha \bar{H}_3 \bar{g}_2 + O(\alpha^2) \tag{16}$$

Now we can consider the case for MER where we define the parameter values as follows extending algorithm 1 where A stands for across batches and W stands for within batches:

$$\theta_0 = \theta_0^A = \theta_{00}^W \tag{17}$$

$$\theta_{01}^W = \theta_{00}^W - \alpha g_{00} \tag{18}$$

$$\theta_{02}^W = \theta_{01}^W - \alpha g_{01} \tag{19}$$

$$\theta_1^A = \theta_0^A + \beta \frac{(\theta_{02}^W - \theta_0^A)}{\alpha} = \theta_0 + \beta \frac{(\theta_{02}^W - \theta_0)}{\alpha} = \theta_{10}^W \tag{20}$$

$$\theta_{11}^W = \theta_{10}^W - \alpha g_{10} \tag{21}$$

$$\theta_{12}^W = \theta_{11}^W - \alpha g_{11} \tag{22}$$

$$\theta_2^A = \theta_1^A + \beta \frac{(\theta_{12}^W - \theta_1^A)}{\alpha} \tag{23}$$

$$\theta = \theta_0^A + \gamma\beta \frac{(\theta_2^A - \theta_0^A)}{\beta} = \theta_0^A + \gamma(\theta_2^A - \theta_0^A) \tag{24}$$

$g_{MER}$ the gradient of Meta-Experience Replay can thus be defined analogously to the gradient of Reptile as:

$$g_{MER} = \frac{\theta_0^A - \theta_2^A}{\beta} = \frac{\theta_0 - \theta_2^A}{\beta} \tag{25}$$

By simply applying Reptile from equation 15 we can derive the value of the parameters $\theta_1^A$ after updating with Reptile within the first batch in terms of the original parameters $\theta_0$:

$$\theta_1^A = \theta_0 - \beta\bar{g}_{00} - \beta\bar{g}_{01} + \beta\alpha\bar{H}_{01}\bar{g}_{00} + O(\beta\alpha^2) \tag{26}$$

By subbing equation 26 into equation 23 we can see that:

$$\theta_2^A = \theta_0 - \beta\bar{g}_{00} - \beta\bar{g}_{01} + \beta\alpha\bar{H}_{01}\bar{g}_{00} - \beta g_{10} - \beta g_{11} + O(\beta\alpha^2) \tag{27}$$

We can express $g_{10}$ in terms of the initial point, by considering a Taylor expansion following the Reptile paper:

$$g_{10} = \bar{g}_{10} + \alpha\bar{H}_{10}(\theta_{10}^W - \theta_0) + O(\alpha^2) \tag{28}$$

Then substituting in for $\theta_{10}^W$ we express $g_{10}$ in terms of $\theta_0$:

$$g_{10} = \bar{g}_{10} - \alpha\beta\bar{H}_{10}\bar{g}_{00} - \alpha\beta\bar{H}_{10}\bar{g}_{01} + O(\alpha^2) \tag{29}$$

We can then rewrite $g_{11}$ by taking a Taylor expansions with respect to $\theta_{10}^W$:

$$g_{11} = g_{11}^{10} - \alpha H_{11}^{10} g_{10} + O(\alpha^2) \tag{30}$$

Taking another Taylor expansion we find that we can transform our expression for the Hessian:

$$H_{11}^{10} = \bar{H}_{11} + O(\alpha) \tag{31}$$

We can analogously also transform our expression our expression for $g_{11}^{10}$:

$$g_{11}^{10} = \bar{g}_{11} + \alpha\bar{H}_{11}(\theta_{10}^W - \theta_0) + O(\alpha^2) \tag{32}$$

Substituting for $\theta_{10}^W$ in terms of $\theta_0$

$$g_{11}^{10} = \bar{g}_{11} - \alpha\beta\bar{H}_{11}\bar{g}_{00} - \alpha\beta\bar{H}_{11}\bar{g}_{01} + O(\alpha^2) \tag{33}$$

We then substitute equation 31, equation 33, and equation 29 into equation 34:

$$g_{11} = \bar{g}_{11} - \alpha\beta\bar{H}_{11}\bar{g}_{00} - \alpha\beta\bar{H}_{11}\bar{g}_{01} - \alpha\bar{H}_{11}\bar{g}_{10} + O(\alpha^2) \tag{34}$$

Finally, we have all of the terms we need to express $\theta_2^A$ and we can then derive an expression for the MER gradient $g_{MER}$:

$$g_{MER} = \bar{g}_{00} + \bar{g}_{01} + \bar{g}_{10} + \bar{g}_{11}$$
$$-\alpha\bar{H}_{01}\bar{g}_{00} - \alpha\bar{H}_{11}\bar{g}_{10} - \alpha\beta\bar{H}_{10}\bar{g}_{00} - \alpha\beta\bar{H}_{10}\bar{g}_{01} - \alpha\beta\bar{H}_{11}\bar{g}_{00} - \alpha\beta\bar{H}_{11}\bar{g}_{01} + O(\alpha^2) \tag{35}$$

This equation is quite interesting and very similar to equation 16. As we would like to approximate the same objective, we can remove one hyperparameter from our model by setting $\beta = 1$. This yields:

$$g_{MER} = \bar{g}_{00} + \bar{g}_{01} + \bar{g}_{10} + \bar{g}_{11}$$
$$-\alpha\bar{H}_{01}\bar{g}_{00} - \alpha\bar{H}_{11}\bar{g}_{10} - \alpha\bar{H}_{10}\bar{g}_{00} - \alpha\bar{H}_{10}\bar{g}_{01} - \alpha\bar{H}_{11}\bar{g}_{00} - \alpha\bar{H}_{11}\bar{g}_{01} + O(\alpha^2) \tag{36}$$

Indeed, with $\beta$ set to equal 1, we have shown that the gradient of MER is the same as one loop of Reptile with a number of steps equal to the total number of examples in all batches of MER (algorithm 6) if the current example is mixed in with the same proportion. If we include in the current example for $s$ of $sk$ examples in our meta-replay batch, it gets the same overall priority in both cases which is $s$ times larger than that of a random example drawn from the buffer. As such, we can also optimize an equivalent gradient using algorithm 7 because it uses a factor $s$ to increase the priority of the gradient given to the current example.

While $\beta = 1$ is an interesting special case of MER in algorithm 1, in general we find it can be useful to set $\beta$ to be a value smaller than 1. In fact, in our experiments we consider the case when $\beta$ is smaller than 1 and $\gamma = 1$. The success of this approach makes sense because the higher order terms in the Taylor expansion that reflect the mismatch between parameters across replay batches disturb the learning process. By setting $\beta$ to a value below 1 we can reduce our comparative weighting on promoting inter batch gradient similarities rather than intra batch gradient similarities.

It was noted in (Nichol & Schulman, 2018) that the following equality holds if the examples and order are random:

$$\mathbb{E}[\bar{H}_1\bar{g}_0] = \mathbb{E}[\bar{H}_0\bar{g}_1] = \frac{1}{2}\mathbb{E}[\frac{\partial}{\partial\theta_0}(\bar{g}_0 \cdot \bar{g}_1)] \tag{37}$$

In our work to make sure this equality holds in an online setting, we must take multiple precautions as noted in the main text. The issue is that examples are received in a non-stationary sequence so when applied in a continual learning setting the order is not totally random or arbitrary as in the original Reptile work. We address this by maintaining our buffer using reservoir sampling, which ensures that any example seen before has a probability $\frac{1}{N}$ of being a particular element in the buffer. We also randomly select over these elements to form a batch. As this makes the order largely arbitrary to the extent that our buffer includes all examples seen, we are approximating the random offline setting from the original Reptile paper. As such we can view the gradients in equation 16 and equation 36 as leading to approximately the following objective function:

$$\theta = arg \min_\theta \mathbb{E}_{(x_{11},y_{11}),...,(x_{sk},y_{sk})\sim M}[2\sum_{i=1}^{s}\sum_{j=1}^{k}[L(x_{ij},y_{ij}) - \sum_{q=1}^{i-1}\sum_{r=1}^{j-1}\alpha\frac{\partial L(x_{ij},y_{ij})}{\partial\theta}\cdot\frac{\partial L(x_{qr},y_{qr})}{\partial\theta}]]. \tag{38}$$

This is precisely equation 7 in the main text.

## J    SUPERVISED CONTINUAL LIFELONG LEARNING

For the supervised continual learning benchmarks leveraging MNIST Rotations and MNIST Permutations, following conventions, we use a two layer MLP architecture for all models with 100 hidden units in each layer. We also model our hyperparameter search after Lopez-Paz & Ranzato (2017) while providing statistics for each model across 5 random seeds.

For Omniglot, following Vinyals et al. (2016) we scale the images to 28x28 and use an architecture that consists of a stack of 4 modules before a fully connected softmax layer. Each module includes a 3x3 convolution with 64 filters, a ReLU non-linearity and 2x2 max-pooling.

## J.1 Hyperparameter Search

Here we report the hyper-parameter grids that we searched over in our experiments. We note in parenthesis the best values for MNIST Rotations (ROT) at each buffer size (ROT-5120, ROT-500, ROT-200), MNIST Permutations (PERM) at each buffer size (PERM-5120, PERM-500, PERM-200), Many Permutations (MANY) at each buffer size (MANY-5120, MANY-500), and Omniglot (OMNI) at each buffer size (OMNI-5120, OMNI-500).

- Online Learning
  - learning rate: [0.0001, 0.0003 (ROT), 0.001, 0.003 (PERM, MANY), 0.01, 0.03, 0.1 (OMNI)]
- Independent Model Per Task
  - learning rate: [0.0001, 0.0003, 0.001, 0.003, 0.01 (ROT, PERM, MANY), 0.03, 0.1]
- Task Specific Input Layer
  - learning rate: [0.0001, 0.0003, 0.001, 0.003, 0.01 (ROT, PERM), 0.03, 0.1]
- EWC
  - learning rate: [0.001 (ROT, OMNI), 0.003 (MANY), 0.01 (PERM), 0.03, 0.1, 0.3, 1.0]
  - regularization: [1 (MANY), 3, 10 (PERM, OMNI), 30, 100 (ROT), 300, 1000, 3000, 10000, 30000]
- GEM
  - learning rate: [0.001, 0.003 (MANY-500), 0.01 (ROT, PERM, OMNI, MANY-5120), 0.03, 0.1, 0.3, 1.0]
  - memory strength ($\gamma$): [0.0 (ROT-500, ROT-200, PERM-200, MANY-5120), 0.1 (MANY-500), 0.5 (OMNI), 1.0 (ROT-5120, PERM-5120, PERM-500)]
- Experience Replay (Algorithm 4)
  - learning rate: [0.00003, 0.0001, 0.0003, 0.001, 0.003, 0.01, 0.03, 0.1 (ROT, PERM, MANY)]
  - batch size ($k$-1): [5 (ROT-500), 10 (ROT-200, PERM-500, PERM-200), 25 (ROT-5120, PERM-5120, MANY), 50, 100, 250]
- Experience Replay (Algorithm 5)
  - learning rate: [0.00003, 0.0001, 0.0003, 0.001, 0.003 (MANY-5120), 0.01 (ROT-500, ROT-200, PERM, MANY-500), 0.03 (ROT-5120), 0.1]
  - batch size ($k$-1): [5 (MANY-500), 10 (PERM-200, MANY-5120), 25 (PERM-5120, PERM-500), 50 (ROT-200), 100 (ROT-5120, ROT-500), 250]
- Meta-Experience Replay (Algorithm 1)
  - learning rate ($\alpha$): [0.01 (OMNI-5120), 0.03 (ROT-5120, PERM, MANY-500), 0.1 (ROT-500, ROT-200, OMNI-500)]
  - across batch meta-learning rate ($\gamma$): 1.0
  - within batch meta-learning rate ($\beta$): [0.01 (ROT-500, ROT-200, MANY-5120), 0.03 (ROT-5120, PERM, MANY-500), 0.1, 0.3, 1.0 (OMNI)]
  - batch size ($k$-1): [5 (MANY, OMNI-500), 10 (ROT-500, ROT-200, PERM-200), 25 (PERM-500, OMNI-5120), 50, 100 (ROT-5120, PERM-5120)]
  - number of batches per example: [1, 2 (OMNI-500), 5 (ROT-200, OMNI-5120), 10 (ROT-5120, ROT-500, PERM, MANY)]
- Meta-Experience Replay (Algorithm 6)
  - learning rate ($\alpha$): [0.01, 0.03 (ROT-5120, PERM-5120, PERM-500, MANY-5120), 0.1 (ROT-500, ROT-200, PERM-200, MANY-500)]
  - meta-learning rate ($\gamma$): [0.03 (ROT-500, ROT-200, PERM-200, MANY-500), 0.1 (ROT-5120, PERM-5120, MANY-5120), 0.3 (PERM-500), 0.6, 1.0]
  - batch size ($k$-1): [5 (PERM-200, MANY-500), 10 (ROT-500, PERM-500) 25 (ROT-200, MANY-5120), 50 (PERM-5120), 100 (ROT-5120), 250]

| Model | FTI |
|---|---|
| Online | $58.22_{\pm 2.03}$ |
| Task Input | $1.62_{\pm 0.87}$ |
| EWC | $58.26_{\pm 1.98}$ |
| GEM | $\mathbf{65.96}_{\pm 1.67}$ |
| MER | $\mathbf{66.74}_{\pm 1.41}$ |

Table 5: Forward transfer and interference (FTI) experiments on MNIST Rotations.

- number of batches per example: 1
- Meta-Experience Replay (Algorithm 7)
  - learning rate ($\alpha$): [0.01 (PERM-5120, PERM-500), 0.03 (ROT, PERM-200, MANY), 0.1]
  - within batch meta-learning rate ($\gamma$): [0.03 (ROT, MANY), 0.1 (PERM), 0.3, 1.0]
  - batch size ($k$-1): [5 (PERM-200, MANY-500), 10, 25 (PERM-500), 50 (ROT-200, ROT-500, MANY-5120), 100 (ROT-5120, PERM-5120)]
  - current example learning rate multiplier ($s$): [1, 2 (PERM-200), 5 (ROT), 10 (PERM-5120, PERM-500, MANY)]

## K  FORWARD TRANSFER AND INTERFERENCE

Forward transfer was a metric defined in Lopez-Paz & Ranzato (2017) based on the average increased performance on a task relative to performance at random initialization before training on that task. Unfortunately, this metric does not make much sense for tasks like MNIST Permutations where inputs are totally uncorrelated across tasks or Omniglot where outputs are totally uncorrelated across tasks. As such, we only provide performance for this metric on MNIST Rotations in Table 5.

## L  ABLATION EXPERIMENTS

We plot our detailed ablation results in Table 6. In order to consider a version of GEM that uses reservoir sampling, we maintain our buffer the same way that we do for experience replay and MER. We consider everything in the buffer to be old data and solve the GEM quadratic program so that the loss is not increased on this data. We found that considering the task level gradient directions did not lead to improvements.

## M  REPRODUCIBILITY OF RESULTS

While the results so far have provided substantial evidence of the benefits of MER for continual learning, one potential concern with our experimental protocol in Appendix J.1 is that the larger hyperparameter search space used for MER may artificially inflate improvements given typical run to run variation. To validate that this is not the case, we have run extensive additional experiments in this section to see how the model performs across different random seeds and machines. The codebase presents some inherent stochasticity across runs. As such, in Tables 7, 8, and 9 we report two levels of generalization for a set of hyperparameters beyond the configuration of an individual run. In the *Same Seeds* column, we report the results for the original 5 model seeds (0-4) deployed on different machines. In the *Different Seeds* column, we report the results for a different 25 model seeds (5-29) also deployed on different machines.

In all cases, we see that there are quantitative differences generalizing across seeds and machines. However, new settings do not always result in lower performance. Additionally, the differences are not qualitative in nature. In fact, in every setting we come to approximately the same qualitative conclusions about how each model performs.

| Model | Buffer Size | Rotations | Permutations | Many Permutations |
|---|---|---|---|---|
| ER with SGD (algorithm 4) | 5120 | $87.82_{\pm 0.44}$ | $84.30_{\pm 0.09}$ | $60.67_{\pm 0.38}$ |
| | 500 | $77.82_{\pm 1.71}$ | $75.80_{\pm 0.54}$ | $44.08_{\pm 0.40}$ |
| | 200 | $70.72_{\pm 1.43}$ | $69.52_{\pm 1.07}$ | — |
| ER with Tasks and SGD (algorithm 5) | 5120 | $88.50_{\pm 1.90}$ | $84.00_{\pm 0.21}$ | $60.05_{\pm 0.24}$ |
| | 500 | $77.30_{\pm 0.42}$ | $74.32_{\pm 0.82}$ | $43.14_{\pm 0.86}$ |
| | 200 | $70.82_{\pm 1.15}$ | $68.06_{\pm 1.14}$ | — |
| MER (algorithm 1) | 5120 | $\mathbf{89.56}_{\pm 0.11}$ | $\mathbf{85.50}_{\pm 0.16}$ | $62.52_{\pm 0.32}$ |
| | 500 | $\mathbf{82.08}_{\pm 0.31}$ | $77.50_{\pm 0.46}$ | $\mathbf{47.40}_{\pm 0.35}$ |
| | 200 | $\mathbf{77.42}_{\pm 0.78}$ | $\mathbf{73.46}_{\pm 0.45}$ | - |
| MER (algorithm 6) | 5120 | $88.94_{\pm 0.17}$ | $84.70_{\pm 0.17}$ | $60.98_{\pm 0.42}$ |
| | 500 | $79.38_{\pm 0.77}$ | $75.88_{\pm 0.19}$ | $44.48_{\pm 0.62}$ |
| | 200 | $73.74_{\pm 1.05}$ | $70.30_{\pm 0.84}$ | - |
| MER (algorithm 7) | 5120 | $89.34_{\pm 0.21}$ | $\mathbf{85.64}_{\pm 0.23}$ | $\mathbf{63.14}_{\pm 0.32}$ |
| | 500 | $\mathbf{82.40}_{\pm 0.64}$ | $\mathbf{78.20}_{\pm 0.57}$ | $46.72_{\pm 0.55}$ |
| | 200 | $\mathbf{77.26}_{\pm 1.19}$ | $\mathbf{73.22}_{\pm 0.37}$ | - |
| ER with Tasks and Adam (Kingma & Ba, 2014) | 5120 | $88.68_{\pm 0.29}$ | $83.78_{\pm 0.19}$ | $58.82_{\pm 0.31}$ |
| | 500 | $77.84_{\pm 0.97}$ | $72.14_{\pm 1.01}$ | $41.10_{\pm 0.24}$ |
| | 200 | $69.48_{\pm 1.21}$ | $63.52_{\pm 0.96}$ | — |
| ER with Tasks and RMSProp (Hinton et al., 2012) | 5120 | $88.28_{\pm 0.16}$ | $82.84_{\pm 0.50}$ | $59.00_{\pm 0.32}$ |
| | 500 | $76.32_{\pm 1.34}$ | $67.80_{\pm 0.80}$ | $36.68_{\pm 0.51}$ |
| | 200 | $66.66_{\pm 0.71}$ | $55.00_{\pm 0.79}$ | — |
| ER with Tasks and GEM Style Buffer | 5120 | $86.78_{\pm 0.37}$ | $81.18_{\pm 0.28}$ | $54.56_{\pm 0.59}$ |
| | 500 | $74.26_{\pm 0.81}$ | $70.04_{\pm 0.48}$ | $38.12_{\pm 0.64}$ |
| | 200 | $66.02_{\pm 0.55}$ | $62.98_{\pm 0.69}$ | - |
| GEM (Lopez-Paz & Ranzato, 2017) | 5120 | $87.58_{\pm 0.32}$ | $83.02_{\pm 0.23}$ | $56.76_{\pm 0.29}$ |
| | 500 | $74.88_{\pm 1.29}$ | $69.26_{\pm 0.66}$ | $32.14_{\pm 0.50}$ |
| | 200 | $67.38_{\pm 0.72}$ | $55.42_{\pm 1.10}$ | - |
| GEM with Reservoir Sampling | 5120 | $87.16_{\pm 0.41}$ | $83.68_{\pm 0.40}$ | $58.94_{\pm 0.53}$ |
| | 500 | $77.26_{\pm 2.09}$ | $74.82_{\pm 0.29}$ | $42.24_{\pm 0.48}$ |
| | 200 | $69.00_{\pm 0.84}$ | $68.90_{\pm 0.71}$ | - |

Table 6: Retained accuracy ablation experiments on MNIST based learning lifelong learning benchmarks.

| Model | Buffer Size | Original | Same Seeds | Different Seeds |
|---|---|---|---|---|
| Online | N/A | $53.38_{\pm 1.53}$ | $53.12_{\pm 1.60}$ | $52.79_{\pm 0.98}$ |
| Independent | N/A | $60.74_{\pm 4.55}$ | $60.64_{\pm 4.21}$ | $66.27_{\pm 5.22}$ |
| EwC | N/A | $57.96_{\pm 1.33}$ | $57.50_{\pm 0.92}$ | $56.30_{\pm 1.74}$ |
| GEM | 5120 | $87.58_{\pm 0.32}$ | $87.56_{\pm 0.36}$ | $87.42_{\pm 0.59}$ |
|  | 500 | $74.88_{\pm 0.93}$ | $73.12_{\pm 1.68}$ | $73.77_{\pm 1.33}$ |
|  | 200 | $67.38_{\pm 1.75}$ | $66.72_{\pm 2.37}$ | $67.45_{\pm 2.01}$ |
| ER with SGD (algorithm 4) | 5120 | $87.82_{\pm 0.44}$ | $87.66_{\pm 0.67}$ | $87.56_{\pm 0.78}$ |
|  | 500 | $77.82_{\pm 1.71}$ | $77.76_{\pm 0.50}$ | $77.48_{\pm 1.57}$ |
|  | 200 | $70.72_{\pm 1.43}$ | $70.50_{\pm 1.36}$ | $70.79_{\pm 1.19}$ |
| ER with Tasks and SGD (algorithm 5) | 5120 | $88.50_{\pm 1.90}$ | $88.14_{\pm 0.70}$ | $88.31_{\pm 0.56}$ |
|  | 500 | $77.30_{\pm 0.42}$ | $77.52_{\pm 0.50}$ | $77.06_{\pm 0.60}$ |
|  | 200 | $70.82_{\pm 1.15}$ | $70.48_{\pm 0.95}$ | $69.55_{\pm 1.09}$ |
| MER (algorithm 1) | 5120 | $\mathbf{89.56}_{\pm 0.11}$ | $\mathbf{89.58}_{\pm 0.12}$ | $\mathbf{89.58}_{\pm 0.19}$ |
|  | 500 | $\mathbf{82.08}_{\pm 0.31}$ | $82.08_{\pm 0.34}$ | $\mathbf{82.21}_{\pm 0.51}$ |
|  | 200 | $\mathbf{77.42}_{\pm 0.79}$ | $77.84_{\pm 1.14}$ | $\mathbf{77.39}_{\pm 0.80}$ |
| MER (algorithm 6) | 5120 | $88.94_{\pm 0.17}$ | $88.80_{\pm 0.17}$ | $88.90_{\pm 0.18}$ |
|  | 500 | $79.38_{\pm 0.77}$ | $79.80_{\pm 0.70}$ | $80.08_{\pm 0.50}$ |
|  | 200 | $73.74_{\pm 1.05}$ | $74.50_{\pm 0.59}$ | $74.32_{\pm 0.91}$ |
| MER (algorithm 7) | 5120 | $89.34_{\pm 0.21}$ | $89.30_{\pm 0.18}$ | $89.33_{\pm 0.13}$ |
|  | 500 | $\mathbf{82.40}_{\pm 0.64}$ | $\mathbf{82.60}_{\pm 0.35}$ | $81.62_{\pm 0.52}$ |
|  | 200 | $\mathbf{77.26}_{\pm 1.19}$ | $\mathbf{77.18}_{\pm 1.25}$ | $\mathbf{76.50}_{\pm 0.94}$ |

Table 7: A reproducability comparison of retained accuracy across machines and seeds for the best performing hyperparameters on MNIST Rotations.

| Model | Buffer Size | Original | Same Seeds | Different Seeds |
|---|---|---|---|---|
| Online | N/A | $55.42_{\pm 0.65}$ | $57.26_{\pm 1.17}$ | $57.45_{\pm 1.25}$ |
| Independent | N/A | $55.80_{\pm 4.79}$ | $55.30_{\pm 4.98}$ | $55.36_{\pm 5.99}$ |
| EwC | N/A | $62.32_{\pm 1.34}$ | $61.40_{\pm 1.89}$ | $60.65_{\pm 2.74}$ |
| GEM | 5120 | $83.02_{\pm 0.23}$ | $82.74_{\pm 0.32}$ | $82.61_{\pm 0.36}$ |
|  | 500 | $69.26_{\pm 0.66}$ | $68.90_{\pm 0.46}$ | $68.38_{\pm 0.87}$ |
|  | 200 | $55.42_{\pm 1.10}$ | $55.48_{\pm 1.37}$ | $54.98_{\pm 1.79}$ |
| ER with SGD (algorithm 4) | 5120 | $84.30_{\pm 0.09}$ | $84.40_{\pm 1.79}$ | $84.40_{\pm 0.29}$ |
|  | 500 | $75.80_{\pm 0.54}$ | $75.90_{\pm 0.49}$ | $75.61_{\pm 0.85}$ |
|  | 200 | $69.52_{\pm 1.07}$ | $69.38_{\pm 1.12}$ | $68.97_{\pm 1.13}$ |
| ER with Tasks and SGD (algorithm 5) | 5120 | $84.00_{\pm 0.21}$ | $83.82_{\pm 0.17}$ | $83.74_{\pm 0.50}$ |
|  | 500 | $74.32_{\pm 0.82}$ | $74.28_{\pm 0.25}$ | $73.72_{\pm 0.70}$ |
|  | 200 | $68.06_{\pm 1.14}$ | $68.06_{\pm 0.21}$ | $67.45_{\pm 1.21}$ |
| MER (algorithm 1) | 5120 | $\mathbf{85.50}_{\pm 0.16}$ | $\mathbf{85.52}_{\pm 0.13}$ | $\mathbf{85.47}_{\pm 0.21}$ |
|  | 500 | $77.50_{\pm 0.46}$ | $\mathbf{77.46}_{\pm 0.38}$ | $\mathbf{77.72}_{\pm 0.44}$ |
|  | 200 | $\mathbf{73.46}_{\pm 0.45}$ | $72.54_{\pm 0.74}$ | $\mathbf{72.65}_{\pm 0.62}$ |
| MER (algorithm 6) | 5120 | $84.70_{\pm 0.17}$ | $84.60_{\pm 0.18}$ | $84.66_{\pm 0.18}$ |
|  | 500 | $75.88_{\pm 0.19}$ | $76.26_{\pm 0.27}$ | $75.65_{\pm 0.86}$ |
|  | 200 | $70.30_{\pm 0.84}$ | $70.12_{\pm 0.92}$ | $69.52_{\pm 0.66}$ |
| MER (algorithm 7) | 5120 | $\mathbf{85.64}_{\pm 0.23}$ | $85.48_{\pm 0.19}$ | $\mathbf{85.58}_{\pm 0.27}$ |
|  | 500 | $\mathbf{78.20}_{\pm 0.57}$ | $\mathbf{77.83}_{\pm 0.55}$ | $\mathbf{77.93}_{\pm 0.63}$ |
|  | 200 | $\mathbf{73.22}_{\pm 0.37}$ | $\mathbf{72.33}_{\pm 1.15}$ | $71.69_{\pm 1.13}$ |

Table 8: A reproducability comparison of retained accuracy across machines and seeds for the best performing hyperparameters on MNIST Permutations.

| Model | Buffer | Original | Same Seeds | Different Seeds |
|---|---|---|---|---|
| Online | N/A | $31.94_{\pm 1.17}$ | $31.92_{\pm 0.96}$ | $31.66_{\pm 1.21}$ |
| Independent | N/A | $13.55_{\pm 3.66}$ | $13.34_{\pm 2.21}$ | $13.42_{\pm 4.49}$ |
| EwC | N/A | $33.46_{\pm 0.46}$ | $32.84_{\pm 0.93}$ | $33.00_{\pm 0.92}$ |
| GEM | 5120 | $56.76_{\pm 0.29}$ | $56.82_{\pm 0.55}$ | $56.68_{\pm 0.51}$ |
|  | 500 | $32.14_{\pm 0.50}$ | $31.86_{\pm 0.64}$ | $31.44_{\pm 0.72}$ |
| ER with SGD (algorithm 4) | 5120 | $60.67_{\pm 0.38}$ | $60.70_{\pm 0.31}$ | $60.64_{\pm 0.35}$ |
|  | 500 | $44.08_{\pm 0.40}$ | $43.54_{\pm 0.57}$ | $43.60_{\pm 0.76}$ |
| ER with Tasks and SGD (algorithm 5) | 5120 | $60.30_{\pm 0.24}$ | $60.24_{\pm 0.32}$ | $60.04_{\pm 0.56}$ |
|  | 500 | $43.14_{\pm 0.86}$ | $43.12_{\pm 0.56}$ | $42.48_{\pm 0.91}$ |
| MER (algorithm 1) | 5120 | $62.52_{\pm 0.33}$ | $62.30_{\pm 0.46}$ | $62.45_{\pm 0.34}$ |
|  | 500 | $\mathbf{47.40}_{\pm 0.35}$ | $\mathbf{47.60}_{\pm 0.62}$ | $\mathbf{47.23}_{\pm 0.68}$ |
| MER (algorithm 6) | 5120 | $60.98_{\pm 0.42}$ | $60.76_{\pm 0.34}$ | $60.94_{\pm 0.31}$ |
|  | 500 | $44.48_{\pm 0.62}$ | $44.28_{\pm 0.42}$ | $44.80_{\pm 0.63}$ |
| MER (algorithm 7) | 5120 | $\mathbf{63.14}_{\pm 0.32}$ | $\mathbf{63.08}_{\pm 0.54}$ | $\mathbf{63.08}_{\pm 0.33}$ |
|  | 500 | $46.72_{\pm 0.55}$ | $46.42_{\pm 0.70}$ | $\mathbf{46.63}_{\pm 0.47}$ |

Table 9: A reproducability comparison of retained accuracy across machines and seeds for the best performing hyperparameters on MNIST Many Permutations.

# N  CONTINUAL REINFORCEMENT LEARNING

We detail the application of MER to deep Q-learning in algorithm 8, using notation from Mnih et al. (2015).

---

**Algorithm 8** Deep Q-learning with Meta-Experience Replay (MER)

---

**procedure** DQN-MER($env, frameLimit, \theta, \alpha, \beta, \gamma, steps, k, E_Q$)
    *// Initialize action-value function Q with parameters $\theta$:*
    $Q \leftarrow Q(\theta)$
    *// Initialize action-value function $\hat{Q}$ with the same parameters $\hat{\theta} = \theta$:*
    $\hat{Q} \leftarrow \hat{Q}(\hat{\theta}) = \hat{Q}(\theta)$
    *// Initialize experience replay buffer:*
    $M \leftarrow \{\}$
    $M.age \leftarrow 0$
    **while** $M.age \leq frameLimit$ **do**
        *// Begin new episode:*
        $env.reset()$
        *// Initialize the s state with the initial observation:*
        **while** episode not done **do**
            *// Select with probability p an action a from set of possible actions:*
            $a = \begin{cases} \text{random selection of action } \hat{a} & p \leq \epsilon \\ \arg\max_{a'} Q(s_t, a'; \theta) & p > \epsilon \end{cases}$
            *// Perform the action a in the environment:*
            $s', r_t \leftarrow env.step(s, a)$
            *// Store current transition with reward r:*
            $M \leftarrow M \cup \{(s, a, r, s')\}$ (algorithm 3)
            $B_1, ..., B_{steps} \leftarrow sample(s, a, r, s', steps, k, M)$
            *// Store current weights:*
            $\theta_0^A \leftarrow \theta$
            **for** $i = 1, ..., steps$ **do**
                $\theta_{i,0}^W \leftarrow \theta$
                **for** $j = 1, ..., k$ **do**
                    *// Sample one set of processed sequences, actions, and rewards from $M$:*
                    $s, a, r, s' = B_i[j]$
                    $y = \begin{cases} r & \text{if final frame in episode} \\ r + \Gamma \max_a \hat{Q}(s', a; \hat{\theta}) & \text{otherwise} \end{cases}$
                    *// Optimize the Huber loss $H(y, Q(s, a; \theta_{i,j-1}^W))$:*
                    $L \leftarrow H(y, Q(s, a; \theta_{i,j-1}^W))$
                    $\theta_{i,j}^W \leftarrow \theta_{i,j-1}^W - \alpha \frac{\partial L}{\partial \theta_{i,j-1}^W}$
                **end for**
                *// Within batch Reptile meta-update:*
                $\theta \leftarrow \theta_{i,0}^W + \beta(\theta_{i,k}^W - \theta_{i,0}^W)$
                $\theta_i^A \leftarrow \theta$
            **end for**
            *// Across batch Reptile meta-update:*
            $\theta \leftarrow \theta_0^A + \gamma(\theta_{steps}^A - \theta_0^A)$
            *// Reset target action-value network $\hat{Q}$ to Q every $E_Q$ number of episodes:*
            $\hat{Q} = Q$
        **end while**
    **end while**
    **return** $\theta, M$
**end procedure**

---

## N.1   DESCRIPTION OF CATCHER AND FLAPPY BIRD

In Catcher, the agent controls a segment that lies horizontally in the bottom of the screen, i.e. a basket, and can move right or left, or stay still. The goal is to move the basket to catch as many pellets as possible. Missing a pellet results in losing one of the three available lives. Pellets emerge one by one from the top of the screen, and have a descending velocity that is fixed for each task.

The reward and thus y axis in our Catcher experiments refers to the number of fruits caught during the full game span.

In the case of the very popular game Flappy Bird, the agent has to navigate a bird in an environment full of pipes by deciding whether to flap or not flap its wings. The pipes appear always in pairs, one from the bottom of the screen and one from the top of the screen, and have a gap that allows the bird to pass through them. Flapping the wings results in the bird ascending, otherwise the bird descends to ground naturally. Both ascending and descending velocities are presets by the physics engine of the game. The goal is to pass through many pairs of pipes as possible without hitting a pipe, as this results in losing the game. The scoring scheme in this game awards a point each time a pipe is crossed. Despite very simple mechanics, Flappy Bird has proven to be challenging for many humans. According to the original game scoring scheme, players with a score of 10 receive a Bronze medal; with 20 points, a Silver medal; 30 results in a Gold medal, and any score better than 40 is rewarded with a Platinum medal.

### N.2 DQN WITH META-EXPERIENCE REPLAY

The DQN used to train on both games follows the classic architecture from Mnih et al. (2015): it has a CNN consisting of 3 layers, the first with 32 filters and an 8x8 kernel, the second layer with 64 filters and a 4x4 kernel, and a final layer with 64 filters and a 3x3 kernel. The CNN is followed by two fully connected layers. A ReLU non-linearity was applied after each layer. We limited the memory buffer size for our models to 50k transitions, which is roughly the proportion of the total memories used in the benchmark setting for our supervised learning tasks.

### N.3 PARAMETERS FOR CONTINUAL REINFORCEMENT LEARNING EXPERIMENTS

For the continual reinforcement learning setting we set the parameters using results from the experiments in the supervised setting as guidance. Both Catcher and Flappy Bird used the same hyper parameters as detailed below with the obvious exception of the game-dependent parameter that defines each task. Models were trained to a maximum of 150k frames and 6 total tasks, switching every 25k frames. Runs used different random seeds for the initialization as stated in the figures.

- Game Parameters
  - Catcher: $\Delta$: 0.03 (vertical velocity of pellet increased from default 0.608).
  - Flappy Bird: $\Delta$: $-5$ (pipe gap decreased 5 from default 100).
- Experience Replay
  - learning rate: 0.0001
  - batch size ($k$-1): 16
- Meta-Experience Replay
  - learning rate ($\alpha$): 0.0001
  - within batch meta-learning rate ($\beta$): 1
  - across batch meta-learning rate ($\gamma$): 0.3
  - batch size ($k$-1): 16
  - number of steps: 1
  - buffer size: 50000

### N.4 FURTHER COMMENTS ON CONTINUAL REINFORCEMENT LEARNING EVALUATION

Performance during training in continual learning for a non-stationary version of Flappy Bird is shown in Figure 5. The graphs show averaged values over three validation episodes across three different seed initializations. Vertical grid lines on the frames axis indicate task switches.

We have also conducted experiments performed using a DQN with reservoir sampling, finding that it consistently underperforms a DQN with typical recency based sampling. A DQN with MER achieves approximately the asymptotic performance for the single task DQN by the end of training for most tasks. On the other hand, DQN with reservoir sampling achieves worse performance than the standard DQN, so it is clear that, in this particular setting where a later task is subsumed in

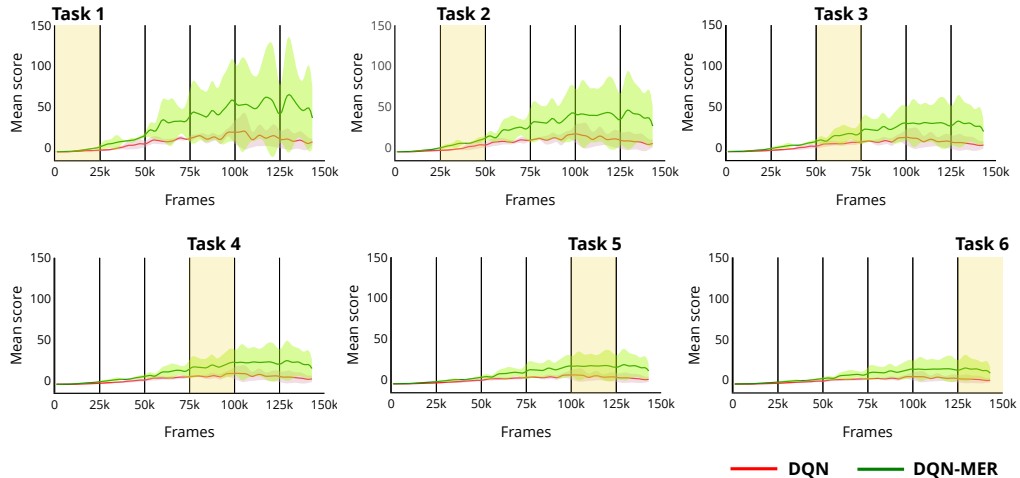

Figure 5: Continual learning for a non-stationary version of Flappy Bird.

previous tasks, keeping easy experiences alone does not account for the benefit of DQN-MER. DQN-MER performs better at training the first task and experiences positive forward transfer for the remaining tasks over what is possible just training for 25k steps on a single task. In most cases, DQN-MER achieves similar performance to the DQN that takes 1 million steps and achieves asymptotic performance. On the first three tasks DQN-MER performs better and it performs a bit worse for the later tasks where it has less time to train. There does not seem to a price paid by DQN-MER in these experiments for not forgetting the easier tasks. Actually, we find that on the final tasks, DQN-MER achieves significant transfer from easier tasks and achieves better performance than the single task DQN does after even training on those tasks alone for 150k steps.

