# OpenReview forum: "Learning to Learn without Forgetting by Maximizing Transfer and Minimizing Interference"
_ICLR.cc/2019/Conference_

### Official Review · AnonReviewer1 · 2018-11-01
**Good paper, more RL experiments and ablations would improve it substantially**

**Rating:** 7
**Confidence:** 5

**Review:**

The paper considers a number of streaming learning settings with various forms of dataset shift/drift of interest for continual learning research, and proposes a novel regularization-based objective enabled by a replay memory managed using the well known reservoir sampling algorithm.

Pros:
The new objective is not too surprising, but figuring out how to effectively implement this objective in a streaming setting is the strong point of this paper.

Task labels are not used, yet performance seems superior to competing methods, many of which use task labels.

Results are good on popular benchmarks, I find the baselines convincing in the supervised case.

Cons:
Despite somewhat frequent usage, I would like to respectfully point out that Permuted MNIST experiments are not very indicative for a majority of desiderata of interest in continual learning, and i.m.h.o. should be used only as a prototyping tool. To pick one issue, such results can be misleading since the benchmark allows for “trivial” solutions which effectively freeze the upper part of the network and only change first (few) layer(s) which “undo” the permutation. This is an artificial type of dataset shift, and is not realistic for the type of continual learning issues which appear even in single task deep reinforcement learning, where policies or value functions represented by the model need to change substantially across learning.

I was pleased to see the RL experiments, which I find more convincing because dataset drifts/shifts are more interesting. Also, such applications of continual learning solutions are attempting to solve a ‘real problem’, or at least something which researchers in that field struggle with. That said, I do have a few suggestions. At first glance, it’s not clear whether anything is learned in the last 3 versions of Catcher, also what the y axis actually means. What is good performance for each game is very specific to your actual settings so I have no reference to compare the scores with. The sequence of games is progressively harder, so it makes sense that scores are lower, but it’s not clear whether your approach impedes learning of new tasks, i.e. what is the price to pay for not forgetting?

This is particularly important for the points you’re trying to make because a large number of competing approaches either saturate the available capacity and memory with the first few tasks, or they faithfully model the recent ones. Any improvement there is worth a lot of attention, given proper comparisons. Even if this approach does not strike the ‘optimal’ balance, it is still worth knowing how much training would be required to reach full single-task performance on each game variant, and what kind of forgetting that induces.

---

> ### Author Response · Authors · 2018-11-26
> **Response to Good paper, more RL experiments and ablations would improve it substantially**
>
> Thank you for your great suggestions about the RL experiments. We have made substantial revisions to the RL experiment sections in the main text and appendix. Additionally, we will still add more ablation experiments that we have performed to our charts for the final draft.
>
> To clarify, the y axis in Catcher refers to the number of fruits caught during the full game span. We have tried to make this more clear within our reinforcement learning experiment details in Appendix M.1.
>
> In the final draft we will provide charts that details the single-task performances after 25k steps, 150k steps and asymptotic performance. For example, we report some of our results for Flappy Bird averaged across runs below to give you an idea of the comparative performance of DQN-MER.
>
> Single Task DQN on Flappy Bird
> =======================
>
> 25k Step Single Task Results:
> DQN Task 0 at 25k steps =-1.13
> DQN Task 1 at 25k steps =-0.47
> DQN Task 2 at 25k steps =-2.66
> DQN Task 3 at 25k steps =-3.95
> DQN Task 4 at 25k steps =-4.14
> DQN Task 5 at 25k steps =-4.95
>
> 150k Step Single Task Results:
> DQN Task 0 at 150k steps = 23.73
> DQN Task 1 at 150k steps = 19.34
> DQN Task 2 at 150k steps = 13.65
> DQN Task 3 at 150k steps = 6.91
> DQN Task 4 at 150k steps = 8.02
> DQN Task 5 at 150k steps = -0.92
>
> 1M Step Single Task Results:
> DQN Task 0 at 1M steps = 28.08
> DQN Task 1 at 1M steps = 25.56
> DQN Task 2 at 1M steps = 17.72
> DQN Task 3 at 1M steps = 17.72
> DQN Task 4 at 1M steps = 14.49
> DQN Task 5 at 1M steps = 10.00
>
> Continual Learning with DQN-MER on Flappy Bird
> =======
>
> Continual Learning Results After 25k Steps On The Task:
> DQN-MER Task 0 after training on Task 0 (at 25k steps) = 1.32
> DQN-MER Task 1 after training on Task 1 (at 50k steps) = 11.92
> DQN-MER Task 2 after training on Task 2 (at 75k steps) = 19.42
> DQN-MER Task 3 after training on Task 3 (at 100k steps) = 21.98
> DQN-MER Task 4 after training on Task 4 (at 125k steps) = 15.30
> DQN-MER Task 5 after training on Task 5 (at 150k steps) = 8.46
>
> Continual Learning Results After Training On All 6 Tasks:
> DQN-MER Task 0 at 150k steps = 36.63
> DQN-MER Task 1 at 150k steps = 26.72
> DQN-MER Task 2 at 150k steps = 19.83
> DQN-MER Task 3 at 150k steps = 14.63
> DQN-MER Task 4 at 150k steps  = 11.06
> DQN-MER Task 5 at 150k steps = 8.46
>
> Clearly DQN-MER performs better at training the first task and experiences positive forward transfer for the remaining tasks over what is possible just training for 25k steps on a single task. In most cases, DQN-MER achieves similar performance to the DQN that takes 1 million steps and achieves asymptotic performance. On the first three tasks DQN-MER performs better and it performs a bit worse for the later tasks where it has less time to train. There does not seem to a price paid by DQN-MER in these experiments for not forgetting the easier tasks. Actually, we find that on the final tasks, DQN-MER achieves significant transfer from easier tasks and achieves better performance than the single task DQN does after even training on those tasks alone for 150k steps.  We have also conducted experiments performed using a DQN with reservoir sampling, finding that it consistently underperforms a DQN with typical recency-based sampling in the RL settings we explore. In the final draft, we will include updated charts with the results of DQN with reservoir sampling and DQN-MER with recency-based sampling added. We really appreciate you suggesting these kinds of experiments and we look forward to improving our charts in the final draft to provide much more context for understanding our RL results.

---

> > ### Comment · AnonReviewer1 · 2018-11-30
> > **Keeping my score**
> >
> > I'm satisfied with the extra information provided by the authors and I'm keeping my score. The improvements suggested by the other reviewers will substantially help the manuscript and should be implemented, but I believe this paper should be accepted.

---

### Official Review · AnonReviewer3 · 2018-11-01
**A promising approach to continual learning that combines experience replay with meta-learning**

**Rating:** 8
**Confidence:** 4

**Review:**

The authors frame continual learning as a meta-learning problem that balances catastrophic forgetting against the capacity to learn new tasks. They propose an algorithm (MER) that combines a meta-learner (Reptile) with experience replay for continual learning. MER is evaluated on variants of MNIST (Permutated, Rotations, Many) and Omniglot against GEM and EWC. It is further tested in two reinforcement learning environments, Catcher and FlappyBird. In all cases, MER exhibits significant gains in terms of average retained accuracy.

Pro's

The paper is well structured and generally well written. The argument is both easy to follow and persuasive. In particular, the proposed framework for trading off catastrophic forgetting against positive transfer is enlightening and should be of interest to the community.

While the idea of aligning gradients across tasks has been proposed before (Lopez-Paz & Ranzato, 2017), the authors make a non-trivial connection to Reptile that allows them to achieve the same goal in a surprisingly simple algorithm. That the algorithm does not require tasks to be identified makes it widely applicable and reported results are convincing.

The authors have taken considerable care to tease out various effects, such as how MER responds to the degree of non-stationarity in the data, as well as the buffer size.  I’m particularly impressed that MER can achieve such high retention rates using only a buffer size of 200. Given that multiple batches are sampled from the buffer for every input from the current task, I’m surprised MER doesn’t suffer from overfitting. How does the train-test accuracy gap change as the buffer size varies?

The paper is further strengthened by empirically verifying that MER indeed does lead to a gradient alignment across tasks, and by an ablation study delineating the contribution from the ER strategy and the contribution from including Reptile. Notably, just using ER outperforms previous methods, and for a sufficient large buffer size, ER is almost equivalent to MER. This is not surprising given that, in practice, the difference between MER and ER is an additional decay rate ( \gamma) applied to gradients from previous batches.

Con's

I would welcome a more thorough ablation study to measure the difference between ER and MER. In particular, how sensitive is MER is to changes in \gamma? And could ER + an adaptive optimizer (e.g. Adam) emulate the effect of \gamma and perform on par with MER. Similarly, given that DQN already uses ER,  it would be valuable to report how a DQN with reservoir sampling performs.

I am not entirely convinced though that MER maximizes for forward transfer. It turns continual learning into multi-task learning and if the new task is sufficiently different from previous tasks, MER’s ability to learn the current task would be impaired. The paper only reports average retained accuracy, so the empirical support for the claim is ambiguous.

The FlappyBird experiment could be improved. As tasks are defined by making the gap between pipes smaller, a good policy for task t is a good policy for task t-1 as well, so the trade-off between backward and forward transfer that motivates MER does not arise. Further, since the baseline DQN never finds a good policy, it is essentially a pseudo-random baseline. I suspect the only reason DQN+MER learns to play the game is because it keeps "easy" experiences with a lot of signal in the buffer for a longer period of time. That both the baseline and MER+DQN seems to unlearn from tasks 5 and 6 suggests further calibration might be needed.

---

> ### Author Response · Authors · 2018-11-26
> **Response to A promising approach to continual learning that combines experience replay with meta-learning**
>
> Thank you for your detailed review and comments about our work.
>
> You bring up an interesting question related to the effect of varying buffer sizes. Based on our experiments, we found that the train-test generalization gap has a complicated relationship with buffer size. The network tends to learn the data that is in the memory buffer at the end of training to approximately perfect accuracy. Intuitively, the network will tend to overfit even more on the buffer data as the buffer becomes smaller. The test set accuracy tends to be higher when the buffer is larger and generalization becomes better as overfitting on the items in the buffer is less of an issue. That being said, the training set accuracy does not necessarily follow the pattern of the accuracy on the items in the buffer. As the model has been potentially trained as little as one step on some examples many training steps ago, models that tend to generalize poorly to the test set also generalize poorly to some parts of the training set that are not included in the buffer.
>
> In order to address your comments about our ablation studies, we have revamped Table 6 of Appendix K to include more experiments to help make our findings clearer. We included, based on your suggestion, experiments demonstrating that adaptive optimizers like Adam and RMSProp do not account for the gap between ER and MER. Particularly for smaller buffer sizes, these approaches seem to overfit more on the buffer and actually hurt generalization in comparison to simple SGD. We also added detail on the performance of the different variants of MER proposed in algorithms 1, 5, and 6. Additionally, we have included new experiments about the impact of the buffer strategy, including those showing how reservoir sampling can also improve GEM although it still slightly underperforms ER.  We have also conducted experiments performed using a DQN with reservoir sampling, finding that it consistently underperforms a DQN with typical recency-based sampling in the RL settings we explore. In the final draft, we will include updated charts with the results of DQN with reservoir sampling and DQN-MER with recency sampling added.
>
> Thank you for your comment about the ambiguity in our experiments. In addition to retained accuracy, we have now also included learned accuracy (LA) which represents the average accuracy for each task directly after learning that task. As you can see in our updated experiments, MER consistently achieves the best performance for this metric as well as retained accuracy. While it is true that attempting to approximate the multi-task setting could potentially result in interference from other tasks, our proposed regularization is seeking to minimize this interference and maximize transfer across tasks which should mitigate the potential for dissimilar tasks to have a negative effect on learning.
>
> We have found that MER, for example in algorithm 6, is not particularly sensitive to the gamma hyperparameter. Overall, for a fixed gamma*alpha which functions as an effective learning rate, we see fairly consistent performance when varying gamma and alpha. In the final draft we will include a chart demonstrating this in the appendix.
>
> We will provide detailed charts in the final draft including performance results for a DQN with reservoir sampling and a single task DQN. Regarding your comments about Flappy Bird, we find that a DQN with MER achieves approximately the asymptotic performance for the single task DQN by the end of training for most tasks. On the other hand, DQN with reservoir sampling achieves worse performance than the standard DQN, so it is clear that, in this particular setting where a later task is subsumed in previous tasks, keeping easy experiences alone does not account for the benefit of DQN-MER.

---

### Official Review · AnonReviewer2 · 2018-11-04
**Nice intuitions on how to think about transfer and interference (thorough rebuttal convinced me to upgrade my rating)**

**Rating:** 6
**Confidence:** 5

**Review:**

The transfer/ interference perspective of lifelong learning is well motivated, and combining the meta-learning literature with the continual learning literature (applying reptile twice), even if seems obvious, wasn't explored before. In addition, this paper shows that a lot of gain can be obtained if one uses more randomized and representative memory (reservoir sampling). However, I'm not entirely convinced with the technical contributions and the analysis provided to support the claims in the paper, good enough for me to accept it in its current form. Please find below my concerns and I'm more than happy to change my mind if the answers are convincing.

Main concerns:

1) The trade-off between transfer and interference, which is one of the main contributions of this paper, has recently been pointed out by [1,2]. GEM[1] talks about it in terms of forward transfer and RWalk[2] in terms of "intransigence". Please clarify how "transfer" is different from these. A clear distinction will strengthen the contribution, otherwise, it seems like the paper talks about the same concepts with different terminologies, which will increase confusion in the literature.

2) Provide intuitions about equations (1) and (2). Also, why is this assumption correct in the case of "incremental learning" where the loss surface itself is changing for new tasks?

3) The paper mentions that the performance for the current task isn't an issue, which to me isn't that obvious as if the evaluation setting is "single-head [2]" then the performance on current task becomes an issue as we move forwards over tasks because of the rigidity of the network to learn new tasks. Please clarify.

4) In eq (4), the second sample (j) is also from the same dataset for which the loss is being minimized. Intuitively it makes sense to not to optimize loss for L(xj, yj) in order to enforce transfer. Please clarify.

5) Since the claim is to improve the "transfer-interference" trade-off, how can we verify this just using accuracy? Any metric to quantify these? What about forgetting and forward transfer measures as discussed in [1,2]. Without these, its hard to say what exactly the algorithm is buying.

6) Why there isn't any result showing MER without reservoir sampling. Also, please comment on the computational efficiency of the method (which is crucial for online learning), as it seems to be very slow.

7)The supervised learning experiments are only shown on the MNIST. Maybe, at least show on CONV-NET/ RESNET (CIFAR etc).

8) It is not clear from where the gains are coming. Do the ablation where instead of using two loops of reptile you use one loop.

Minor:
=======
1) In the abstract, please clarify what you mean by "future gradient". Is it gradient over "unseen" task, or "unseen" data point of the same task. It's clear after reading the manuscript, but takes a while to reach that stage.
2) Please clarify the difference between stationary and non-stationary distribution, or at least cite a paper with the proper definition.
3) Please define the problem precisely. Like a mathematical problem definition is missing which makes it hard to follow the paper. Clarify the evaluation setting (multi/single head etc [2])
4) No citation provided for "reservoir sampling" which is an important ingredient of this entire algorithm.
5) Please mention appendix sections as well when referred to appendix.
6) Provide citations for "meta-learning" in section 1.


[1] GEM: Gradient episodic memory for continual learning, NIPS17.
[2] RWalk: Riemannian walk for incremental learning: Understanding forgetting and intransigence, ECCV2018.

---

> ### Author Response · Authors · 2018-11-26
> **Part 1 of Response to Nice intuitions on how to think about transfer and interference, but not good enough technical contributions**
>
> Thank you for your detailed review and questions. We will address each comment individually:
>
> Main Concern #1) Thank you for pointing out that the terminology used in our submitted version may be confusing. As you pointed out, it is important to make clear that many of the main ideas we used in our paper including the concepts of transfer and interference in forward and backward directions, the link between transfer and weight sharing, and the idea of involving gradient alignment in a formulation for continual learning have been explored before. The main contribution of the transfer-interference tradeoff we propose in this work is a novel perspective on the goal of gradient alignment for the continual learning problem. We have added additional details in the abstract, Section 1, Section 2, and Appendix B in an attempt to make the comparative novelty of our approach clearer. The transfer-interference tradeoff view of continual learning can be very useful as this temporally symmetric view of this tradeoff in relation to weight sharing leads to a natural meta-learning perspective of continual learning. We have attempted to make this clearer in Figure 1 and Section 2 Footnote 3. Moreover, we have added Appendix C to make the connection with weight sharing more explicit.
>
> However, our operational measures of transfer and interference are in fact the same as forward and backward transfer considered in (Lopez-Paz & Ranzato, NIPS 2017). Following the terminology of (Lopez-Paz & Ranzato, NIPS 2017), we simply use the term “transfer” to refer to our temporally symmetric view of the problem that does not make a distinction between the forward and backward direction. We use “interference” as is common in the literature to refer to the case where transfer is negative. Intransigence and forgetting are also very related to our work as well as the stability-plasticity dilemma. Intransigence and forgetting measure very similar phenomenon to the metrics learned accuracy (LA) and backward transfer and interference (BTI) that we have added to our experiments. We should clarify that we do not consider the way we measure performance to be novel or noteworthy. We have tried to emphasize this by adding additional performance measures such as backward transfer (BTI) and forward transfer (FTI) as used in (Lopez-Paz & Ranzato, NIPS 2017) to our experiments.
>
> Main Concern #2) We have tried to make it clear at the beginning of Section 2 that these operational statements only hold at an instant in time with a set of parameters theta. Because we are considering both data points to be evaluated by the same set of parameters, these equations hold despite the fact that the data points may be drawn from different tasks. This is in fact very similar to the instantaneous notion of transfer considered for continual learning in (Lopez-Paz & Ranzato, NIPS 2017) with the main distinction being that we consider transfer on the example level and not the task level. Obviously, you are right that gradients with respect to the parameters at different points in time may be out of date, which would mean these equations wouldn’t hold. However, it is important to note that we do not implement this case even in the continual learning setting as replayed memories are always considered with the current parameters theta along with the current example. It is true that the notion of generalizing based on this learning about transfer and interference into the future will itself be a non-stationary learning problem. This is because as the parameters change, the notion of good updates for transfer and interference with past examples changes as well. That being said, we are also stabilizing learning for this non-stationary process with experience replay.
>
> Main Concern #3) Thank you for your comment. We would first like to clarify that our experiments on Omniglot would be considered “multi-head” (Chaudhry et al., ECCV 2018). We have updated the text to make this clearer. We have also added a new metric learned accuracy (LA) representing performance on a task right after learning that task to our supervised learning experiments and made the task switches clearer for our RL experiments to directly address your concern. Empirically speaking we find that MER results in the best LA in all cases. Despite using a single head, MER is apparently able to efficiently navigate the transfer-interference tradeoff of weight sharing to achieve good LA while at the same time achieving good backward transfer and interference (BTI) performance.

---

> > ### Author Response · Authors · 2018-11-26
> > **Part 2 of Response to Nice intuitions on how to think about transfer and interference, but not good enough technical contributions**
> >
> > Main Concern #4) We are not sure that we totally follow your intuition. When you consider the effective loss function being optimized over in the offline case (which we are discussing in Equation 4) the extra L(xj,yj) term really only has the effect of increasing the priority of the traditional supervised learning loss function rather than the regularization term. This effect should be largely arbitrary because it can be absorbed by tuning alpha. We have edited the text to further emphasize this point. We report it in this way because this is consistent with what we do in our implementation.
> >
> > Main Concern #5) Thank you for this suggestion as this really improves our discourse. Following (Lopez-Paz & Ranzato, NIPS 2017) in addition to retained accuracy, we now also report backward transfer / interference (BTI) and forward transfer / interference (FTI). Unfortunately, forward transfer only makes sense for single headed settings with correlated tasks, which only applies to our MNIST-Rotations experiments. We include these results in Table 5 of Appendix K. As such, we report the accuracy on a task directly after learning that task (LA) for all of our experiments to express plasticity to incoming tasks. We can see in all cases that the high retained accuracy achieved by MER is the byproduct of the best balance between learned accuracy (LA) and backward transfer / interference (BTI).
> >
> > Main Concerns #6 and #7) In order to address your question about getting rid of reservoir sampling, we have added experiments using the buffer strategy from (Lopez-Paz & Ranzato, NIPS 2017) instead to our ablation experiments in Table 6 of Appendix L. Our experiments demonstrate that reservoir sampling results in the best performance for all methods. ER and GEM perform similarly regardless of the buffer management policy. We have preliminary results for MER without using reservoir sampling as well which we will include in the final draft. Regardless of buffer strategy, MER results in considerable improvements on top of both ER and GEM, especially for small buffer sizes. Thank you for mentioning the computational efficiency of MER. In Figure 2 we highlight the performance characteristics on Omniglot for which we use CNN models in a supervised learning setting. We highlight that MER achieves clearly the best tradeoff between learning performance and computation time as methods like GEM have a difficult time scaling to this kind of architecture. We have worked to make it clearer in the text that we use CNNs here in addition to in our RL DQN experiments.
> >
> > Main Concern #8) Thank you for your comment. We have proposed three variants of MER in this work which we detail in algorithms 1,5, and 6 in the updated draft. What you are asking for with one straightforward Reptile loop is detailed in algorithm 5, where algorithms 1 and 6 provide different mechanisms of adding more weight to the current example. We provide results for all variants of these models and not just algorithm 1 in Table 6 of Appendix L and provide more detail about the connection between the different approaches in Appendix H and I.  We summarize these results in the second paragraph related to Question 6 in Section 6 of the main text. Algorithm 5 results in significant gains over ER and GEM in all cases. Additionally, algorithms 1 and 6 result in further gains on top of that by increasing the prioritization of the current example.
> >
> > Minor Comment #1) Thank you for pointing out the possible confusion here. We have added Footnote 1 to the abstract in order to help clarify this confusion at the onset of the paper. In this work, we focus on algorithms that are agnostic to task boundaries, so we really mean both gradients with respect to unseen examples of the current task and gradients with respect to unseen examples of unseen tasks.
> >
> > Minor Comments #2 and #3) Thank you for the comment. This is a good point. We have added Appendix A to make our definition of the problem and nonstationary setting more rigorous.
> >
> > Minor Comment #4) Thank you for bringing our attention to this issue. We now provide a comprehensive overview of reservoir sampling in Appendix F and algorithm 3.
> >
> > Minor Comments #5 and #6) Thank you for these suggestions. We have addressed them in the revised submission.

---

> > > ### Comment · AnonReviewer2 · 2018-12-03
> > > **Updated review**
> > >
> > > Thank you for your thorough reply. I'm satisfied with the updated draft, it's much cleaner and easy to follow.  Most of my comments have been addressed and incorporated in the updated draft. I am upgrading my rating.

---

### Comment · AnonReviewer1 · 2018-10-26
**Nice work! Could you please provide some extra details?**

Thanks for the paper! I'm particularly impressed by the RL experiments, which I find a bit difficult to fully interpret without more information. For example:
- What are the maximum scores achievable in these games/versions?
- What score does DQN get asymptotically on each version separately, and how much data is required?
- How much can be learned in 25K frames from scratch in each game?
- How does DQN perform with reservoir sampling without MER? Any ablation experiments and data would be useful.
- What is the asymptotic effect of MER on a single task. Does it get to the same level of performance as DQN with enough data? Is this the case for all tasks considered?

---

> ### Author Response · Authors · 2018-10-31
> **Thank you for your comment! Some additional details:**
>
> We would like to sincerely thank you for your comments about our work and for your questions. These will help us further improve our empirical discourse. We will definitely make sure that we address all of your questions in the revised version of our paper once the open review tool allows for revisions.
>
> First Question: Thank you for bringing this important detail up. The answer requires some contextualization. In the case of Catcher, there is no predefined (hard coded) maximum score in the library we used. Under some soft assumptions but with a realistic settings, such as the defaults used in the experimental section (default pellet speed, default player speed, etc), the score grows approximately linearly with the number of frames for a perfect player. It can be approximated by 0.12 x n_frames (empirically we found it could be possible to achieve a score of 1 in 10 frames, 12 in 100 frames, 120 in 1k frames, 597 in 5k frames, and so on). In the case of FlappyBird,  based on reported videos on popular channels, the hard limit of the original game was set to 999 points. However, for the emulator used in this experiment there is no trace of such a hard limit. Maybe a more interesting question is human performance: the fact that it was a very popular game raised the public question of the overall difficulty of the game for humans (see https://www.theguardian.com/news/2014/mar/03/flappy-bird-what-does-the-data-say). As it is stated in the article referenced above, and even in the Wikipedia article, human performance is on average much lower than this hard limit: in the analysis above it is observed that it typically takes more than 350 attempts (full episodes) to achieve a couple of games with score 12. It makes sense to us then that a 'Platinum level' is achieved with a score of 40. We are compiling more information to reliably compute the distribution of scores in human players and we will update the appendix of the revision with this information.
>
> Second and Fifth Questions: The question of asymptotic scores for the RL experiments is an interesting one. We are running experiments now and think this is a good suggestion that will help provide additional context for the results. As a sneak peek for soft reference, our preliminary experiments with another model (A3C) resulted in 296 as the asymptotic score for Catcher. We have found that learning may proceed quite slowly after the initial period, so we would like to run our models for a very long period to ensure we have truly found the asymptotic performance.
>
> Third Question: Thank you for this question as it also improves our discourse to highlight this point, showcasing the significant extent of transfer across tasks that MER achieves during continual lifelong training. We originally provided this information through our figures in the main text, but will make sure to update the format of the figures and provide details in the text to make this much clearer. After 25k steps of training from scratch, a DQN achieves an average score across runs of 143.02 on Catcher and -2.83 on Flappy Bird. In contrast, MER achieves an average score across runs of 187.93 on Catcher and 1.32 on Flappy Bird.
>
> Fourth Question: Thank you for suggesting this ablation experiment. It fits nicely in the context of our ablation analysis section. This will help highlight the value add of incorporating meta-learning.

---

### Meta-Review · Area_Chair1 · 2018-12-14

**Confidence:** 5
**Recommendation:** Accept (Poster)

**Metareview:**

Pros:
- novel method for continual learning
- clear, well written
- good results
- no need for identified tasks
- detailed rebuttal, new results in revision

Cons:
- experiments could be on more realistic/challenging domains

The reviewers agree that the paper should be accepted.